# Improving early warning of drought-driven food insecurity in Southern Africa using operational hydrological monitoring and forecasting products

Shraddhanand Shukla[1], Kristi R. Arsenault[2,3], Abheera Hazra[4,3], Christa Peters-Lidard[3], Randal D. Koster[3], Frank Davenport[1], Tamuka Magadzire[5,1], Chris Funk[6,1], Sujay Kumar[3], Amy McNally[2,5], Augusto Getirana[4,3], Greg Husak[1], Ben Zaitchik[7], Jim Verdin[8,5], Faka Dieudonne Nsadisa[9], Inbal Becker-Reshef[3,4]

[1]University of California, Santa Barbara, California, USA

[2]SAIC, Reston, Virginia, USA

[3]NASA Goddard Space Flight Center, Greenbelt, Maryland, USA

[4]University of Maryland, Maryland, USA

[5]Famine Early Warning Systems Network, Washington D.C., USA

[6]EROS, United States Geological Survey, Sioux Falls, South Dakota, USA

[7]John Hopkins University, Baltimore, Maryland, USA

[8]United States Agency for International Development, Washington D.C., USA

[9]Southern African Development Community Climate Services Center, Botswana

*Correspondence to*: Shraddhanand Shukla (sshukla@ucsb.edu)

**Abstract:**

The region of southern Africa (SA) has a fragile food economy and is vulnerable to frequent
droughts. Interventions to mitigate food insecurity impacts require early warning of droughts —
preferably as early as possible before the harvest season (typically, starting in April) and lean
season (typically, starting in November). Hydrologic monitoring and forecasting systems provide
a unique opportunity to support early warning efforts, since they can provide regular updates on
available rootzone soil moisture (RZSM), a critical variable for crop yield, and provide forecasts
of RZSM by combining the estimates of antecedent soil moisture conditions with climate
forecasts. For SA, this study documents the predictive capabilities of RZSM products from a
recently developed NASA Hydrological Forecasting and Analysis System (NHyFAS). Results
show that the NHyFAS products would have identified the regional severe drought event—
which peaked during December-February of 2015/2016—at least as early as November 1, 2015.
Next, it is shown that during 1982-2016, February RZSM forecasts [monitoring product]
available in early November [early March] have a correlation of 0.49 [0.79] with the detrended
regional crop yield. It is also found that when the February RZSM forecast [monitoring product]
available in early November [early March] is indicated to be in the lowest tercile, the detrended
regional crop yield is below normal about two-thirds of the time [always], at least over the
sample years considered. Additionally, it is shown that February RZSM forecast [monitoring
product] can provide "out-of-sample" crop yield forecasts with comparable [substantially better
with 40% reduction in mean error] skill to December-February ENSO. These results indicate that
the NHyFAS products can effectively support food insecurity early warning in the SA region.
Finally, since a framework similar to NHyFAS can be used to provide RZSM monitoring and
forecasting products over other regions of the globe, this case study also demonstrates potential
for supporting food insecurity early warning globally.

# 1 Introduction

Southern Africa (SA) is vulnerable to food insecurity. Droughts driven by climate stressors (e.g. precipitation and temperature) are among the important drivers of food insecurity (Misselhorn 2005; Conway et al. 2015). Moreover, anthropogenic climate change is shown to increase the likelihood of climate-driven flash droughts (Yuan et al., 2018). The primary rainy season in SA spans from October to March, which overlaps the main planting season from October to February (Fig. 1 [a]). This period also covers the lean season, when food supplies from the prior year's harvest become limited. April-July is typically the main harvest season, when the food reserve is expected to begin replenishing. In several SA countries, with the Republic of South Africa (RSA) being the main exception, typical monthly variability in food prices closely follows this crop cycle, as shown in Fig. 1(b). The prices typically start to rise after the harvest season and reach their peak just before or near the start of the harvest season. This correspondence between the prices and crop cycles highlights the region's climate-related sensitivity to food insecurity. In the case of below-normal crop yield, the food prices rise even more than normal, reducing access to food for the poorest of the population.

The percentage income shared by the poorest 10% and 20% of the population in several SA countries has not improved significantly over time (not shown here). These portions of the population are likely to be more food insecure in drought years; they already use a relatively higher share of their income on food, and in the case of price rises related to low crop yield, their access to food becomes even more limited.

The 2015-16 drought event (attributed to a strong El Niño) in SA further highlighted its vulnerability to climate-related regional food insecurity (Archer et al., 2017; Funk et al., 2018; Pomposi et al., 2018). This event led to a substantial reduction in regional agricultural production

—including in the RSA, which is the main crop-producing country in the region—a reduction
and rationing of water supplies, a loss of livestock, and an increase in unemployment in the
region, and it pushed 29 million people into severe food insecurity (SADC, 2016). Throughout
the Southern African Development Community (SADC) region in 2015-16, cereal production
was down by -10.2% (varying from +61% to -94% in individual member countries) relative to
the previous 5-year average (SADC, 2016). Figure 1 (c)-(f) shows a comparison of national retail
maize prices (in USD) in several of the SA countries during 2015-16, with the previous 5-year
mean prices in those countries. The prices in 2015-16 were substantially higher than the previous
5-year mean. Of particular importance is the price increase in RSA, where, typically, the food
prices do not vary much throughout the year due to its general self-sufficiency in food
production, as well as its international trade. Consumer Price Index (CPI) for food for the RSA
also experienced a drastic upward shift during the 2015-16 drought year (not shown here). In
fact, based on the CPI data (available from the FAO), the CPI was substantially higher than that
of the past 5-year mean during the beginning of the following growing season of 2016-17,
including in the RSA where typically the CPI remains fairly stable during a year. These price
shocks can dramatically impact poor households, which typically spend 60% or more of their
income on food. According to the recent World Development Indicator (World Bank 2016),
incomes for the poorest 10% and 20% of households in these countries have remained generally
constant, underscoring the depth of poverty (Figure 2). On average, in Malawi, Mozambique,
Zimbabwe, and South Africa, these individuals subsist on USD 70, 126, 288, and 716 a year,
respectively.

Figure 1(c)-(f) and the income-related facts (based on World Bank Development

Indicator) presented above highlight the severity of food insecurity in a regional drought event
like 2015-16. In the 2015-2016 event, food imports from the RSA—which is the main producer
and exporter of food in the region to the other SA countries—were not enough, and international
assistance became crucial. This is why in June 2016, the SADC launched a Regional
Humanitarian Appeal stating that approximately 40 million people in the region required
humanitarian assistance, at a cost of approximately USD 2.4 billion (Magadzire et al. 2017).

Mitigation of the most adverse impacts of food insecurity, like the event of 2015-16,

requires timely and effective early warning. An effective early warning system has two key
attributes (Funk et al., 2019): (1) the ability to provide routine, frequent early warning of drought
status and (2) the ability to incorporate both monitoring and forecasting to best account for the
conditions up to the date of early warning, in combination with the climate outlook for the
upcoming season.

A seasonal-scale hydrologic forecasting system can potentially support an early warning

system, as it can provide updated hydrologic forecasts on a monthly basis by accounting for the
drought conditions as of the forecast release date and climate outlook over the forecast period
(Sheffield et al., 2014; Shukla et al., 2014; Yuan et al., 2013). However, thus far, the application
of seasonal-scale hydrologic forecasts in food insecurity early warning has been limited at best,
with the only other main example being the African Flood and Drought Monitor (Sheffield et al.,

2014).

On the other hand, operational, publicly available, state-of-the-art dynamical climate

forecasts have found regular usage in guiding climate outlooks, as well as assessments of
expected food insecurity. For example, USAID's Famine Early Warning Systems Network
(http://fews.net/), G20-Group on Earth Observations Global Agricultural Monitoring
(GEOGLAM) Crop Monitor for Early Warning, and SADC's Climate Service Center (CSC) all
utilize the dynamical climate forecasts as one of their early warning tools. Furthermore,
numerous past studies have investigated the predictability of SA climate (Meque and Abiodun,
2014) and examined the skill of diverse approaches in forecasting, particularly of rainfall, as well
as streamflow and agricultural production in different parts of this region (Archer et al., 2017;
Cane et al., 1994; Diro, 2015; Landman et al., 2001; Landman and Beraki, 2010; Landman and
Goddard, 2002; Manatsa et al., 2015; Martin et al., 2000; Sunday et al., 2014; Trambauer et al.,
2015; Winsemius et al., 2014). Historically, El Niño-Southern Oscillation (ENSO) has proven to
be among the main predictors of this region's climate, with another important predictor being the
Southern Indian Ocean Dipole (Hoell et al., 2016, 2017; Hoell and Cheng, 2017).

In August 2018, a new NASA Hydrological Forecasting and Analysis System

(NHyFAS), an operational seasonal hydrologic forecasting system (Arsenault et al., 2020),  was
implemented to support the early warning efforts of FEWS NET, building upon existing
hydrologic monitoring (McNally et al., 2017). This study evaluates this system's ability to
support early warning of regional food insecurity in the SA region. The evaluation is conducted
by examining the performance of this system (i) for the 2015-16 drought event, which led to
regional food insecurity, (ii) in explaining regional crop yield variability in the region, and (iii) in
identifying below-normal crop yield events, which are characteristically associated with overall
lower food availability in the region and, hence, food insecurity. Regional crop yield is used as a
target variable here, as it is among the main contributors to regional food insecurity. It is
hypothesized that if this system can skillfully forecast regional crop yield and identify below-
normal regional crop yields, it can successfully support the early warning of food insecurity in
the region.
As noted above and shown in Fig. 1(a), April-July is typically the main harvest season,
when the food reserve is expected to begin replenishing and last through the lean season, which
starts in November. Below-normal food availability during this period can lead to food
insecurity. Therefore, early warning systems aim to provide outlooks for food insecurity as far in
advance of the harvest and lean season as possible. Consequently, this study focuses on using
forecasting and monitoring products that are available in November (4-5 months before the start
of the harvest, and about a year before the start of the next lean season) through March (1-2
months before the start of the harvest, and about 8-9 months before the start of the next lean
season) to examine their value in supporting early warning of food insecurity in the region.
**2 Data and Methodology**
The hydrologic monitoring and forecasting products used in this study come from the
NHyFAS (Fig. 3) (Arsenault et al., 2020). Figure 3 shows an overview of the implementation of
the NHyFAS for the purpose of this study. Because Arsenault et al. (2020) already describes the
system in detail, we simply provide here a brief description of the hydrologic models (section
2.1), the model parameters (section 2.2), the input observed forcings and climate forecasts
(section 2.3), and the RZSM monitoring and forecasting products (section 2.4) used in the
present study. The reported crop yield data used in this study are described in section 2.5.
**2.1 Hydrologic Modeling Framework**
To generate hydrological forecasts, we use NASA's Catchment land surface model
(CLSM; (Ducharne et al., 2000; Koster et al., 2000) and the Noah Multi-Parameterization (Noah-
MP; (Niu et al., 2011; Yang et al., 2011) land surface model (LSM), which compute changes in
soil moisture (e.g., root zone) and groundwater storage in response to computed  surface energy
and water fluxes. These two LSMs are part of the model suite in the Land Information System
(LIS) framework (Kumar et al., 2006)—the primary software system used to produce this study's
forecast experiments. Both LSMs were spun-up using two cycles of forcing for the period from 1
January, 1981 to 31 December, 2015; then, historical open-loop (OL) runs were generated for
January 1981 through 2018. Rootzone SM (RZSM), which is the main hydrologic variable used
in this analysis, represents the soil moisture in the top one meter of the soil profile. The entire
depth of the soil profile is different for the two models used in this analysis (typically about 2 m
for Noah-MP, and about 4 m for CLSM).
**2.2 Model Parameters**
In the version of CLSM used here, hydrologic and catchment parameters (Ducharne et
al., 2000) are based on a high-resolution, global topographic data set (Verdin and Verdin, 1999),
and soil texture (Reynolds et al., 2000) and profile parameters are derived from the Second
Global Soil Wetness Project (GSWP-2; Guo and Dirmeyer, 2006) data set and mapped to the
catchment tiles. Land cover classes are mapped from the University of Maryland AVHRR data
set, and vegetation parameters include, for example, leaf area index (LAI), which is also derived
from GSWP-2. Albedo scaling factors are based on Moderate Resolution Imaging
Spectroradiometer (MODIS) direct and diffuse visible or near infra-red radiation inputs (Moody
et al., 2008).
Noah-MP vegetation parameters include the modified IGBP MODIS-based land cover
data set (Friedl et al., 2002), leaf area index, and monthly greenness fraction (Gutman and
Ignatov, 1998). The soil texture data set is based on Reynolds et al. (2000), and soil parameters
are mapped to the varying textures. Monthly global (snow-free) albedo (Csiszar and Gutman,
1999) and a maximum snow albedo parameter field are also employed. Additional details are
found in (Niu et al., 2011).

**2.3 Input observed forcings and climate forecasts**

The spin-up and OL runs used to generate the long-term "observed" climatology of

RZSM are driven with NASA's Modern-Era Retrospective analysis for Research and

Applications, version 2 (MERRA-2; [Gelaro et al., 2017]) atmospheric fields (e.g., 2m air

temperature, humidity). Precipitation forcing comes from the U.S. Geological Survey

(USGS)/University of California, Santa Barbara (UCSB) Climate Hazards Center InfraRed

Precipitation with Station data set, version 2.0 (CHIRPSv2; [Funk et al., 2015]).

Hindcasts of RZSM are generated by forcing the hydrologic models with NASA's

Goddard Earth Observing System (GEOS) Atmosphere-Ocean General Circulation Model,

version 5 (GEOS; [Borovikov et al., 2017]) Seasonal-to-Interannual Forecast System. The eleven

ensemble members of version 1 of this forecast system that were used in the North American

Multi-Model Ensemble (NMME) project are used in the forecast portion of this study. To make

the GEOS forecasted meteorology consistent with the meteorology underlying the OL initial

conditions, we Bias-Corrected and Spatially Downscaled (BCSD; [Wood et al., 2002]) the

GEOS forecasts using the MERRA-2 and CHIRPS data sets. The BCSD-GEOS forecast files are

then ingested into LIS to drive the LSMs and generate the dynamical hydrological forecasts. The

BCSD-GEOS hindcasts are initialized on November 1st (near the start of the planting season)

and January 1st (middle of the planting season) of each year in 1982-83 to 2017-18. Each

hindcast is run for 6 months.

Hindcasts of RZSM are also generated using the Ensemble Streamflow Prediction (ESP)

method (Day 1985; Shukla et al. 2013), where the models are forced with resampled observed

forcings (forcings that are used to drive the OL simulation) taken from 1982-2010 period. The

hindcasts generated using the ESP method derive their skills from the initial hydrologic
conditions only.

**2.4 RZSM Monitoring and forecasting products**

The performance of the NHyFAS system is evaluated mainly through its RZSM
monitoring (generated from OL) and forecasting products. RZSM indicates the soil moisture in
the top one meter of the soil profile. Typically, the length of the roots of crops such as maize
(main crop in the region of SA) is close to one meter, hence the choice of RZSM as the key
forecast variable. Moreover, the entire depth of the soil profile is different for the two models
used in this analysis, typically about 2 m for Noah-MP and about 4 m for CLSM; hence RZSM
also allows for a consistent way to merge soil moisture products from both models.
Both products are generated at 0.25 X 0.25 degree spatial resolution and daily temporal
resolution. Daily values are averaged over a month to get monthly values. The monthly values of
the monitoring product are converted to percentiles relative to OL climatology over 1982-2010,
and monthly values of the ensemble mean forecasting products (GEOS and ESP based) are
converted into percentiles relative to the (ensemble mean) climatology over 1982-2010 of the
respective hindcast runs. In both cases, empirical distribution is considered to convert values to
percentiles. Once gridded percentile values are generated, they are spatially aggregated over the
SA region (as shown in Fig. 2) to get RZSM monitoring and forecasting products over the SA
region.

**2.5 Regional Crop Yield**

The regional crop yield is calculated using country-level crop production and area
harvested reports. These reports come from the United States Department of Agriculture's
Foreign Agricultural Service's Production Supply and Distribution (PSD) database. To compile
this database, USDA relies on several sources, including official country statistics, reports from
agricultural attaches at U.S. embassies, data from international organizations, publications from
individual countries, and information from traders both inside and outside of the target countries.
For this study, we focus only on maize, as it is the main crop in the region and the key crop for
food security. To get regional crop yield from country-level crop yield, we first converted
country-level yield into production using the harvested area (provided by the PSD), added the
total production, and then divided it by the sum of the harvested area in all SA countries in our
focus domain. The regional crop yield is detrended for the purposes of this study to reduce the
effect of any long-term changes (e.g. technological changes) on the crop yield.

**2.6 Out-of-sample crop yield forecasting**
We also evaluate the NHyFAS RZSM monitoring and forecasting products' performance
in supporting food insecurity early warning in SA through a series of out-of-sample crop
forecasting experiments. Specifically, we compare the accuracy of crop yield forecasts made
with NHyFAS products against univariate yield forecasts (using only the past yields) and yield
forecasts made with ENSO, a widely used predictor for crop yield in this region. This evaluation
has a direct implication on the usage of NHyFAS products for operational purposes, as crop yield
forecasts are a common tool in food security analysis and response (Davenport et al., 2019).
Our baseline model is a univariate (no exogenous predictors) Autoregressive Integrated
Moving Average (ARIMA) model,

$$y'_t = \phi_1 y'_{t-1} + \cdots + \phi_p y'_{t-p} + \theta_1 \varepsilon_{t-1} + \cdots + \theta_q \varepsilon_{t-q} + \varepsilon_t, \qquad (1)$$

Where $y_t$ is the time series of observed yields (and the ` indicates potential differencing of the
time series), $p$ is the order of lags, $\phi$ are the autoregressive parameters, $q$ is the order of moving
averages, $\theta$ are the moving average parameters, and $\epsilon$ are forecast errors from the prior periods.
ARIMA(p,d,q) models are standard and frequently used methods for time series analysis and
forecasting (Hyndman and Athanasopoulos, 2018; Hyndman and Khandakar, 2007). As
discussed above, we compare the forecast performance of univariate ARIMA models eq.[1], with
ARIMA models that also include environmental exogenous predictors, which, in this case, are (i)
DJF ENSO (ii) February RZSM monitoring product and (iii) February RZSM forecast initialized
on Nov. 1, during the growing season preceding harvested yields in year $t$ (e.g. 1982/83 DJF
used for 1983 yield). All models are fit using the auto.arima() function from the forecast
package in the R software language.

We use the period of 1983-2007 (25 years) as a training period and then provide "out-of-

sample" forecasts of crop yield starting in 2008. The training period always spans through the
year before the target forecast year. For example, the model fit over 1983-2008 is used to
forecast yields in 2009, and the model fit over 1983-2009 is used to forecast yield in 2010, and
so on. We repeat this exercise through 2018 and record the one-step-ahead prediction error in
each iteration. In this way, we emulate the forecasting process that food security analysts in the
region go through during every year prior to harvest.

**3. Results**
**3.1 Performance of NHyFAS during the 2015-16 drought event**

As highlighted in section 1, the 2015-16 drought event in SA is among the most severe in

terms of drought severity and food insecurity impacts in the last few decades. Therefore, we
begin the evaluation of the suitability of NHyFAS in supporting food insecurity early warning in
the SA region by examining how this system would have performed during the 2015-16 event.
Although the NHyFAS operationally provides the seasonal forecasts every month, for the
purpose of this study, we focus on the forecast initialized on November 1 (near the start of the
planting season) and January 1 (near the middle of the growing season) of 2015-16 event. Figure
4 shows the RZSM forecasts for the growing season made on November 1, 2015. By this time in
the season, both FEWS NET and SADC had provided early warning of poor rainfall
performance in the region (Magadzire et al, 2017). The NHyFAS RZSM forecasts would have
provided further evidence of a looming unprecedented drought in the region. These forecasts
would have also indicated that RSA, which is the most important country for the region's food
production, was going to be within the epicenter of this drought event. These forecasts, in turn,
could potentially have triggered earlier appropriate actions by the early warning agencies, as well
as the decision-makers (e.g., national governments and international relief agencies).
Later in the season, as the observed precipitation data became available, RZSM
monitoring products would have provided refined estimates of the spatial extent and severity of
drought in the region. Figure 4 (bottom panel) shows the RZSM monitoring product available
after each of the months of November 2015 through February 2016. This monitoring product
would have provided additional proof of the drought occurrence in the region, and shown that
RSA was within the epicenter of this drought. It is important to state that even the monitoring
product can be effectively used as a predictor of food insecurity events, as they are available
before the typical start of the harvest season (in April) and the lean season (in November).
**3.2    Performance of NHyFAS in supporting food insecurity early warning**
Next, we investigate the long-term performance of NHyFAS in supporting food
insecurity early warning by examining how well forecasting and monitoring products available
from this system can explain historical variability in regional crop yield of the SA region and in
particular, help identify below-normal regional yield events. Regional crop yield is calculated by
adding the yearly productions from the SA countries, then dividing it by the yearly total
harvested area. The regional crop yield is then detrended to remove the effect of any long-term
changes (such as technological changes) on the regional yield.
First, we show in Figure 5 how detrended crop yield correlates (from early November to
early March) with the monthly RZSM monitoring product relative to how it correlates with 3-
monthly seasonal precipitation and air temperature. The results indicate that the monthly RZSM
monitoring product generally correlates better with detrended crop yield than with the seasonal
precipitation or air temperature, with the correlation reaching its peak by early March, when the
Feb-RZSM monitoring product and December-February precipitation and temperature are
available. Feb-RZSM still shows higher correlation than seasonal precipitation and temperature;
however, the difference in correlation is not statistically significant.
Next, the correlation between detrended crop yield and February RZSM forecasts (based
on ESP method and bias-corrected GEOS forecasts) initialized on November 1 (Fig. 6a) and
January 1 (Fig. 6b) is analyzed. The correlation of the yield with GEOS-based February RZSM
forecasts initialized on November 1 is 0.49, which is substantially higher than that of ESP-based
RZSM forecasts (0.16), clearly demonstrating the added value of using GEOS-based climate
forecasts. Similarly, the correlation of yield with the GEOS-based February RZSM forecasts
initialized on January 1 is 0.45, higher than that of the ESP-based forecasts (0.30) at that time of
the year. Moreover, the correlation of detrended crop yield with GEOS-based February RZSM
forecasts initialized on November 1 (0.49) and January 1 (0.45) is higher than that with the
RZSM monitoring product (Figure 5) at those times of the year (<0.1 in early November and
<0.4 in early January). Again, this highlights the value of using forecasts of Feb-RZSM through
early January in supporting food insecurity early warning. Figure 6c shows that Feb-RZSM
monitoring product, which is available in early March, has the highest correlation of 0.79 with
the detrended crop yield.

Next, we examine how well the forecasting and monitoring RZSM products do in

providing early warning of below-normal crop yield events. This criterion for performance
evaluation is of particular significance for food insecurity early warning in the region, as below-
normal crop yield events are the ones that generally lead to food insecurity. In this case, below-
normal regional crop yield events are the events that lie in the bottom 18 (i.e. bottom half) when
detrended crop yields for the 36 years are ranked in ascending order.

We calculate the probability of below-normal crop yield events when either the February

RZSM forecast (initialized on November 1 and January 1) or the RZSM monitoring product for
the month of November (available in early December) through the month of February (available
in early March) is in the lowest tercile. RZSM products in this tercile are those lying in the
bottom 12 of the RZSM products when ranked in ascending order. In the case of RZSM, the
ranked climatology is different for each of the forecasting products and the monitoring products
for each month. We use the lower tercile values of RZSM monitoring and forecasting products to
focus on the drought years as indicated by those products. Because SA is a mostly rainfed region,
the crop yield is generally below normal during drought years, as indicated in several recent
events (2014-15, 2015-16, 2018-19).

Figure 7 shows the fraction of years with below-normal crop yield when February RZSM

forecasts (made on November 1 or January 1) were in the lower tercile (shown by blue color
bars) or when monthly RZSM monitoring products (shown by green color bars) were in the
lower tercile. These results indicate that as early as November 1, if the February RZSM is being
forecasted to be in the lower tercile, then there is about ~66% probability of the regional crop
yield being below normal (statistically significant at 86% confidence level). This would be 4-5
months before the start of the harvest season, and about one year before the start of the next lean
season. The inferred probability value increases to ~83% when the February RZSM forecasts,
initialized in January, are in the lower tercile (statistically significant >95% confidence level).
Finally, by early March, when the February RZSM monitoring product is available, the inferred
probability increases to 100% (statistically significant >95% confidence level). In other words,
over 1982-2016, whenever the February RZSM monitoring product for the SA region was in the
lowest tercile, the crop yield in the following season had been below normal (based on detrended
yield). This would be 1-2 months before the start of the harvest season, and about 8-9 months
before the start of the next lean season.
Of course, the estimation of these probabilities is necessarily limited by the small sample
sizes examined; the actual probability of low crop yield based on low February RZSM, for
example, while apparently high, is not a full 100%. Nevertheless, these results provide, overall,
further evidence of the suitability of the forecasting and monitoring products from the NHyFAS
in supporting early warning of food insecurity in the region.

**3.3     Performance of NHyFAS in providing routine operational crop yield forecasts**
Finally, we evaluate the performance of NHyFAS for supporting food insecurity early
warning in SA by examining the accuracy of RZSM monitoring and RZSM forecasting products
in predicting regional crop yields. We compare the crop yield forecasts made with the RSZM
products against both univariate forecasts (using only past observed crop yields) and forecasts
made with ENSO. As ENSO is a widely used predictor for precipitation and crop yield forecasts
in this region, we examine the added value of using NHyFAS RZSM monitoring and forecasting
products above and beyond ENSO. All forecasts are done using ARIMA models described in
section 2.6.

Figure 8 shows a comparison between the "observed" reported crop yield (black lines)

and the "out-of-sample" (i.e. post-training period) forecasted yield (red lines) produced with a
univariate model, and the models using environmental exogenous predictors (i) DJF ENSO, (ii)
Feb-RZSM (monitoring) product, (iii) Feb-RZSM (Forecasting product) initialized on Nov. 1., in
addition to that univariate model.
The results indicate that: (i) environmental predictors such as ENSO and the NHyFAS products
can make crop yield forecasts that are more accurate than those produced using only a univariate
approach. When ENSO is used as an additional predictor (in addition to a Univariate model), the
MAE reduces from 0.342 MT/HA to 0.285 MT/HA, a ~17% reduction in error. (ii) Use of the
Feb-RZSM monitoring product has an even larger impact, reducing the MAE by about 50%, to
0.174 MT/HA. (iii) Use of the Feb-RZSM forecasting product (initialized on Nov 1) has an
impact similar to that of DJF ENSO.  Although the MAE is about 6% larger when the forecasting
product is used rather than the ENSO predictor, the forecasting product has the significant
advantage of being available for about 4 months earlier. For comparison (not shown here) MAE
of Feb-RZSM forecasting product (initialized on Nov 1) is slightly smaller (~6%) than the MAE
of August-October (ASO)-ENSO (also available in early Nov) and is comparable to the MAE of
September-November (SON)-ENSO (available in early December) as a predictor of crop yield
forecast.

Table 1 shows the number of times the observed yield is within the 80% confidence

interval of the forecasts and the mean spread of the confidence interval. The improvement in
performance obtained when the Feb-RZSM monitoring product is used is clear; during 10 of the
11 years in the validation period, the observed yield falls within the 80% confidence interval,
whereas this happens in only 7 years when DJF ENSO is used as the additional predictor. The
mean spread of the confidence interval associated with the use of the Feb-RZSM monitoring
product (0.70 MT/HA) is also the smallest.
**4        Discussion**

This study makes a case for the application of NHyFAS's RZSM forecasting and

monitoring products in supporting the early warning of food insecurity in SA. It has been shown
that the successful early warning of crop yield, and especially below-normal crop yield years,
can be issued based on these products. In this section, we address a few important caveats.

**4.1        Comparison with existing drought forecasting systems and approaches:**

In this study, we keep the comparison with existing forecasting systems and approaches

limited to the comparison of the performance of NHyFAS products with (i) ESP (i.e.
climatology) based RZSM forecasts and (ii) ENSO-based crop yield forecasts, both of which are
commonly used approaches for drought forecasting in the region, including by early warning
agencies such as FEWS NET. Comparison against both approaches shows clear added value of
using the NHyFAS products. We could not compare the performance of the NHyFAS with
FEWS NET or SADC's official historical forecasts because:
(i) FEWS NET's official forecast is an outlook of food insecurity conditions (Funk et al. 2019)
(https://fews.net/) which is based not only on agroclimatology (i.e., agriculture and climate
conditions) but also on market conditions and nutrition and livelihood conditions. The NHyFAS
forecasts that are now being used by FEWS NET would fall into the category of
agroclimatological conditions. In fact, the goal of the evaluation of the NHyFAS forecasts is to
establish whether NHyFAS forecasts can be suitable agroclimatological forecast inputs for
FEWS NET to guide the development of food insecurity outlook assessments. Also, FEWS NET
Food Insecurity Outlook is partly based on subjective assessments, in some ways similar to the
U.S. drought monitor (Svoboda et al., 2002) or U.S. Seasonal Drought Outlook, in addition to
quantitative assessments such as agroclimatological forecasts. Finally, FEWS NET's archive of
Food Insecurity Outlooks currently extends back only to mid-2011.
(ii) SADC CSC's issues probabilistic seasonal-scale rainfall forecasts. These forecasts are based
on multiple models (both statistical and dynamical) as well as subjective expert assessments,
which makes comparison with purely quantitative products inappropriate. Additionally, the
archive of purely quantitative forecasts from SADC CSC only goes back to 2017.
Finally, the NHyFAS products are intended to be used as an addition to the existing early
warning tools of FEWS NET and SADC CSC, which are partners in the efforts described in this
study, rather than replacing any of the existing tools.

**4.2    Influence of crop yield on regional food insecurity and issues in crop yield reports**
In this study, it is assumed that when the SA region faces a production shortfall, the
regional food insecurity is likely to rise. This was certainly the case during the 2015-16 El Niño,
the most recent major food insecurity event in the region (SADC 2016). However, this
assumption ignores other important factors that may lead to or further worsen food insecurity in
the region, such as inadequate agricultural inputs, price shocks (which can be global in nature),
rise in population, conflict, limited livelihood options, stocks, etc. Nonetheless, the direct
relationship of crop yield with the interannual variability in available moisture makes RZSM an
important variable for food security monitoring and thus, it is of keen interest to early warning
systems like FEWS NET, which is presently the primary end user of the NHyFAS. Crop yield
early warning based on the NHyFAS products are also directly relevant to international
collaborative efforts like the GEOGLAM initiative (Becker-Reshef et al. 2018; Becker-Reshef et
al. 2019) and, particularly, to the Crop Monitor for Early Warning (https://cropmonitor.org/),
which provides monthly assessments of crop conditions for the countries most vulnerable to food
insecurity. Such assessments are key to reducing the uncertainty of crop prospects as the growing
season progresses, and to providing critical evidence for informing food security decisions by
humanitarian organizations and governments alike.

It is also worth noting that crop yield reports can be influenced by external factors (for

example, reporting issues related to methods) other than long-term agricultural, technology-
driven changes and climate interannual variability. The effect of these factors on the regional
crop yield, of course, cannot be discounted by the detrending method employed in this study.
**4.3     Reliance on single climate model forecasts:**

Finally, the results of this study are also likely affected by the use of only one dynamical

climate forecast model for driving the seasonal hydrologic forecasting system. Adding forecasts
from more climate and hydrologic models would likely enhance the skill of the system (Kirtman
et al. 2014; Krishnamurti et al. 1999). The choice of one dynamical system was made mostly for
logistical purposes, since GEOS archived and real-time forecasts include all atmospheric forcing
variables needed to drive such LSMs, and are available through NASA-GSFC routinely, to
facilitate operational production of NHyFAS forecasts.
**5      Conclusions**
The region of SA witnessed several severe food insecurity events in the last few decades.
Mitigation of food insecurity impact requires timely and effective interventions by national,
regional, and international agencies. To support those interventions, early warning of food
insecurity is needed. In this study, we investigate the suitability of the operational RZSM
products produced by a recently developed NASA seasonal scale hydrologic forecasting system,
NHyFAS, in supporting food insecurity early warning in this region.
The key findings of this study are: (i) the NHyFAS products would have identified the
regional severe 2015-2016 drought event (which peaked in December-February) at least as early
as November 1st of 2015; (ii) February RZSM forecasts produced as early as November 1 (4-5
months before the start of harvest, and about one year before the start of the next lean season)
can explain the interannual variability in regional crop yield production with moderate skill
(correlation 0.49); (iii) use of dynamical climate forecasts adds to the skill (relative to the skill
coming from the initial hydrologic conditions alone) in predicting regional crop yield through the
prediction of February RZSM; (iv) the February RZSM monitoring product, available in early
March (1-2 months before the start of harvest and 8-9 months before the start of the next lean
season) can explain the variability in regional crop yield with high skill (correlation of 0.79); (v)
when the February RZSM forecast (initialized on November 1) is found to be in the lowest
tercile, the subsequent detrended regional crop yield is below normal about 66% of the time
(statistical significance level ~86%), and likewise, when the February RZSM monitoring product
is in the lowest tercile, the subsequent crop yield is (for a limited set of samples considered)
always below normal (statistical significance level >95%); (vi) the February RZSM monitoring
product can provide "out-of-sample" crop yield forecasts with higher skill than DJF ENSO (38%
reduction in mean error relative to DJF ENSO), whereas the February RZSM forecasting
product, available in early November, can provide crop yield forecasts with comparable skill
(~6% increase in mean error relative to DJF ENSO).

The NHyFAS products described here were first generated in August 2018 for

operational applications by FEWS NET. As described in much detail in Funk et al., (2019), each
month, FEWS NET's regional scientists (located in eastern, western, and southern Africa)
review the latest products ahead of the FEWS NET's monthly climate discussions. The NHyFAS
products, in addition to other early warning tools, are used to support or revise the assumptions
of climate and hydrologic conditions in the upcoming season. The updated assumptions are then
passed on to food analysts for the region in order to help inform needed relief actions. This study
demonstrates the value of the NHyFAS products in supporting food insecurity early warning in
the SA region. It is worth mentioning that since NHyFAS currently covers Africa and the Middle
East region, the NHyFAS products are applicable for food insecurity early warning in the rest of
Africa and the Middle East as well. Based on this study, it is postulated (future research pending)
that NHyFAS RZSM products can be particularly effective for those rainfed agriculture regions
and seasons which are not known to have strong teleconnection (e.g. with ENSO), as in the SA
region. Finally, since the data sets and models used to impelement the NHyFAS are available
globally, a similar seasonal RZSM monitoring and forecasting framework can be developed at a
global scale to support food insecurity early warning in other rainfed regions across the globe.

**Author contribution:** SS led the design of the analysis, conducted the analysis, and wrote the
manuscript and generated figures. KA, CPL, CF, and FD contributed to the design of the
analysis. FD contributed to the analysis as well. KA and AH conducted the model simulations.
RK and CPL reviewed the article and proposed substantial changes. CPL and GH are PIs of
projects supporting this work. TM, JV, AM, AH facilitate real-time application of the products.
The other co-authors reviewed the article and provided their input/edits.
**Competing interests:**
The authors declare that they have no conflict of interest.





















**Acknowledgements:**
Support for this study comes from NASA Grant NNX15AL46G and the US Geological Survey
(USGS) cooperative agreement #G09AC000001. Crop yield, production, and consumption data
were obtained from USDA FAS's PSD:
https://apps.fas.usda.gov/psdonline/app/index.html#/app/home. Average price data were obtained
from FAO's FAO STATS database http://www.fao.org/faostat/en/#home. World Bank
Development Indicators were downloaded from https://data.worldbank.org/indicator/. GEOS
forecast data sets are generated and supported by NASA's Global Modeling and Assimilation
Office (GMAO). Model source code can be found at NASA's Land Information System's
GitHub repository (https://lis.gsfc.nasa.gov/news/latest-lis-code-now-available-github). Model
parameters are available through email request. The daily CHIRPS precipitation data can be
found here (ftp://ftp.chg.ucsb.edu/pub/org/chg/products/CHIRPS-2.0/global_daily/netcdf/p25/).
MERRA-2 reanalysis-based atmospheric forcings can be found through NASA's GES DISC
archive (https://disc.gsfc.nasa.gov/datasets?keywords=%22MERRA-
2%22&page=1&source=Models%2FAnalyses%20MERRA-2).  NHyFAS forecasts, in the form
of maps, can be found here https://lis.gsfc.nasa.gov/projects/nhyfas. As of now, NHyFAS
forecast data sets are not publicly accessible. High-performance computing resources were
provided by the NASA Center for Climate Simulation (NCCS) in Greenbelt, MD.  The authors
thank Climate Hazards Center's technical writer, Juliet Way-Henthorne, for providing
professional editing.

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

Table 1: Performance of 'out-of-sample' crop yield forecasting over the validation period of

2008-2018.


|  | Univariate model | Univariate model + ENSO | Univariate model + Feb-RZSM (Monitoring) | Univariate model + Feb-RZSM (forecast) |
|---|---|---|---|---|
| **Mean absolute error over the validation period (MT/HA)** | 0.342 | 0.285 | 0.174 | 0.301 |
| **Number of years observed yield is within 95% confidence interval bound** | 9 | 10 | 10 | 9 |
| **Mean spread of 95% confidence interval (MT/HA)** | 1.64 | 1.20 | 1.07 | 1.20 |
| **Number of years observed yield is within 80% confidence interval bound** | 9 | 7 | 10 | 7 |
| **Mean spread of 80% confidence interval (MT/HA)** | 1.07 | 0.78 | 0.70 | 0.78 |


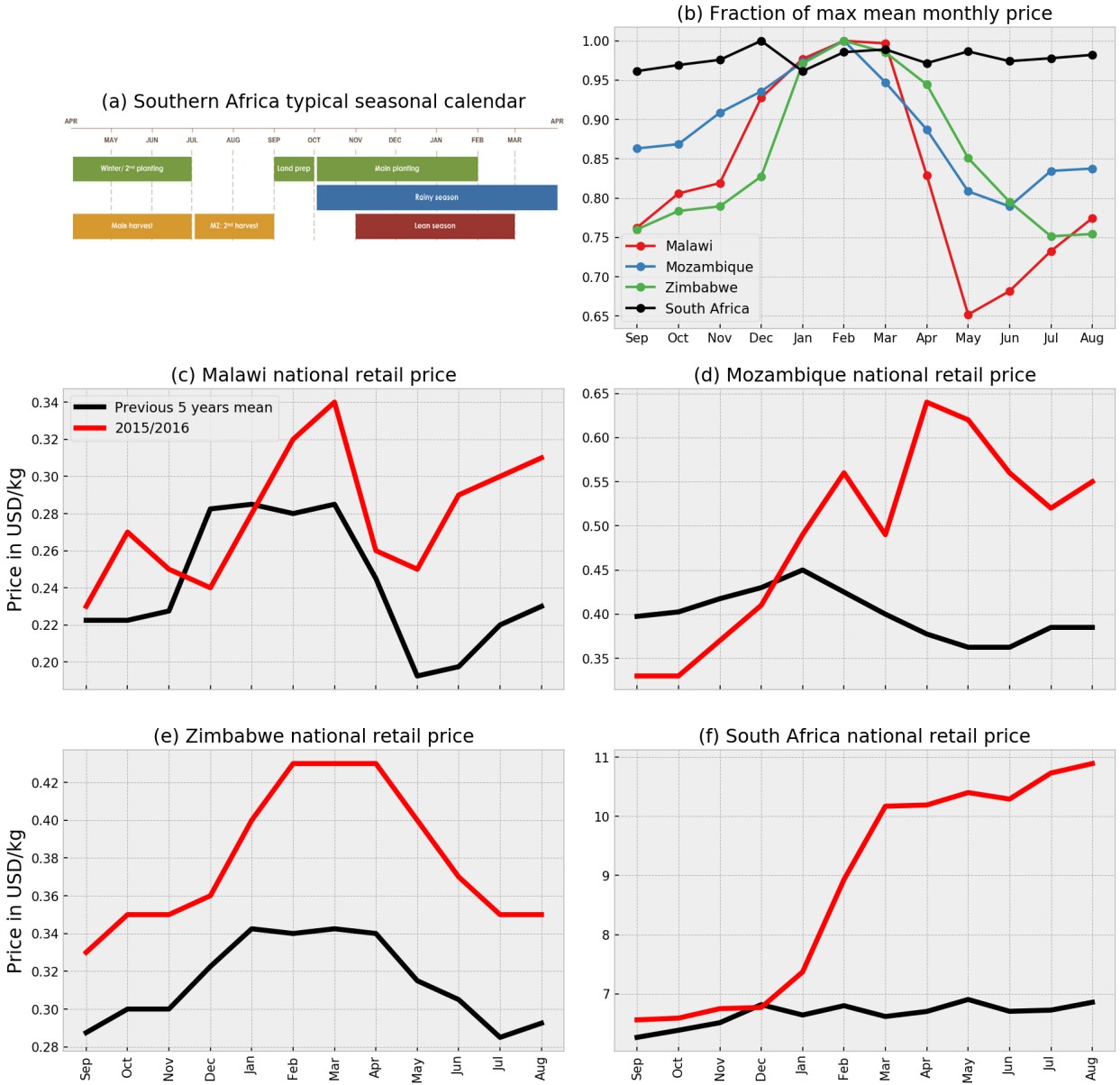



**Figure 1: (a) Schematic representation of a typical seasonal calendar for the southern Africa region. (taken from: http://fews.net/southern-africa)  (b) Monthly climatology of maize prices in SA countries. The monthly mean prices are normalized relative to the maximum mean monthly price for a given country, as the actual values of the mean monthly prices are different for different countries. Comparison of mean monthly maize**

**prices for (c) Malawi (d) Mozambique (e) Zimbabwe (f) South Africa, during the 2015-16**
**event (red line) with the previous 5-year mean prices (black line). The price data is**
**available from FAOSTAT (FAO 2019).**

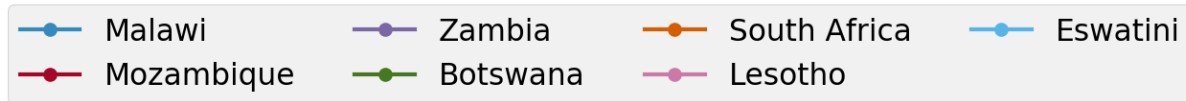

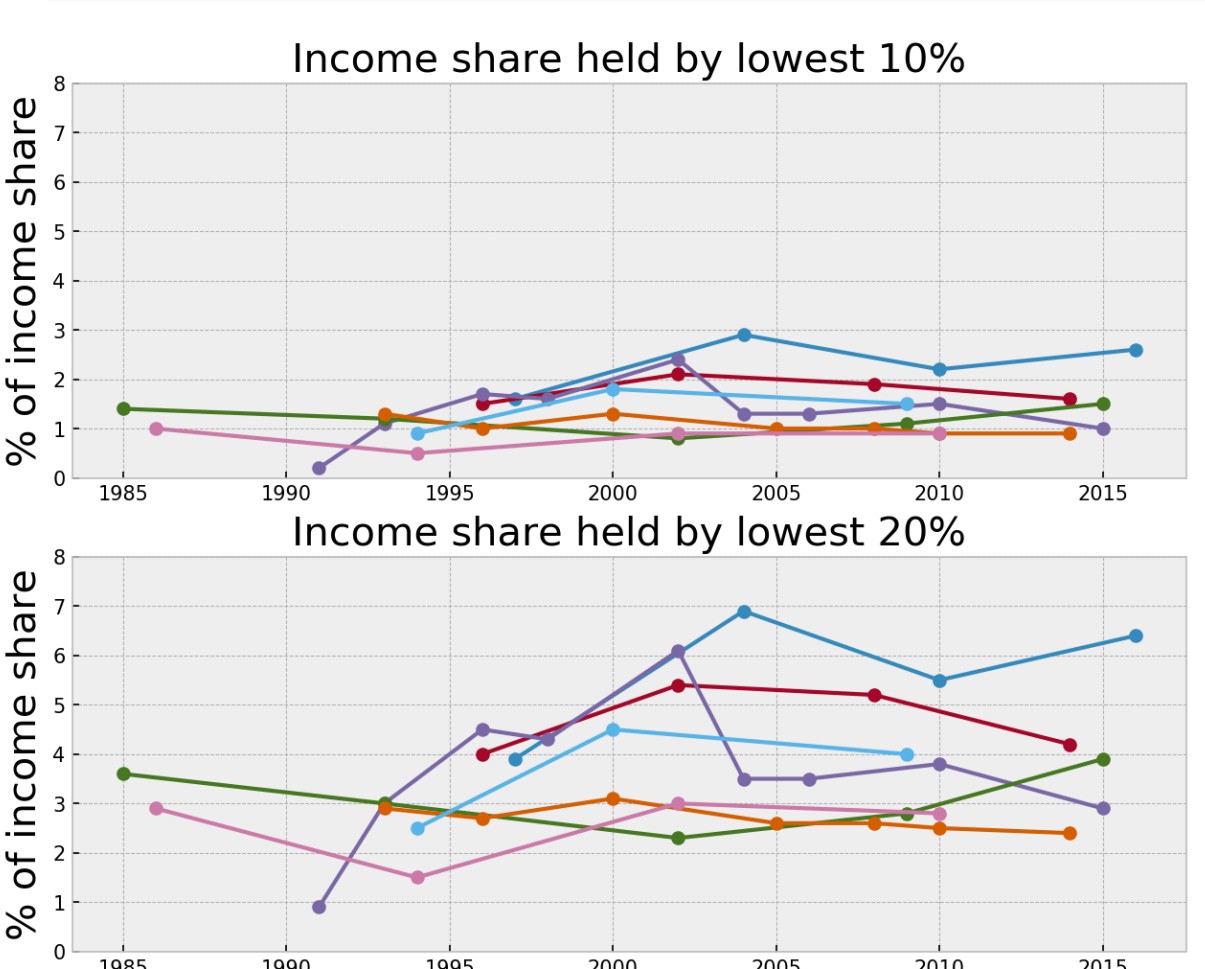

**Figure 2: Percentage of income share held by lowest 10% and 20% income population in the Southern Africa countries. (Data Source: the World Bank's World Development Indicators)**

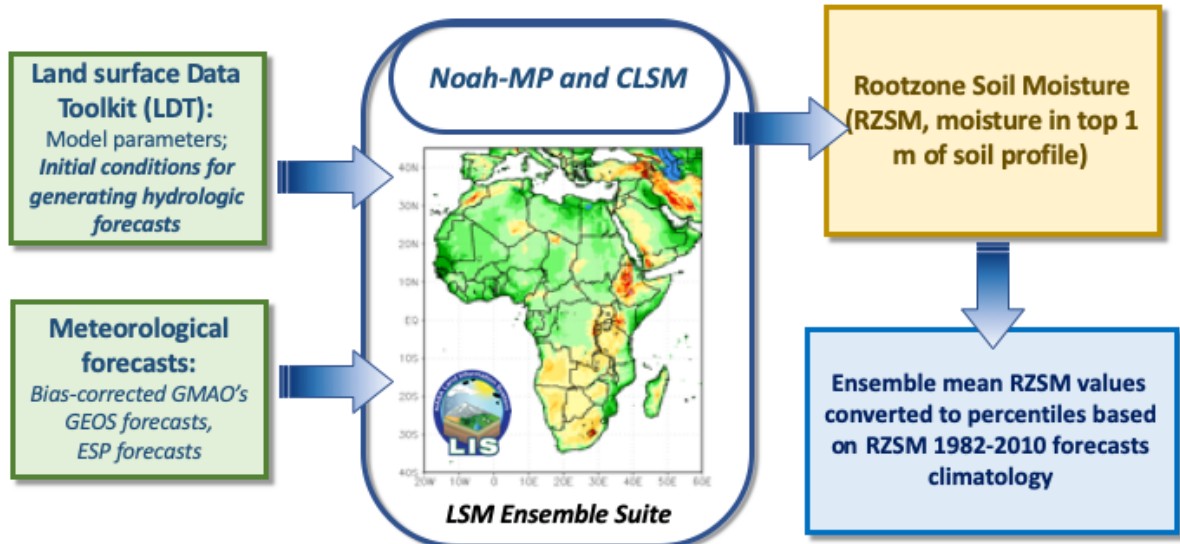


**Figure 3: Overview of the NHyFAS implementation to produce RZSM monitoring and forecasting products, as used in this study.**






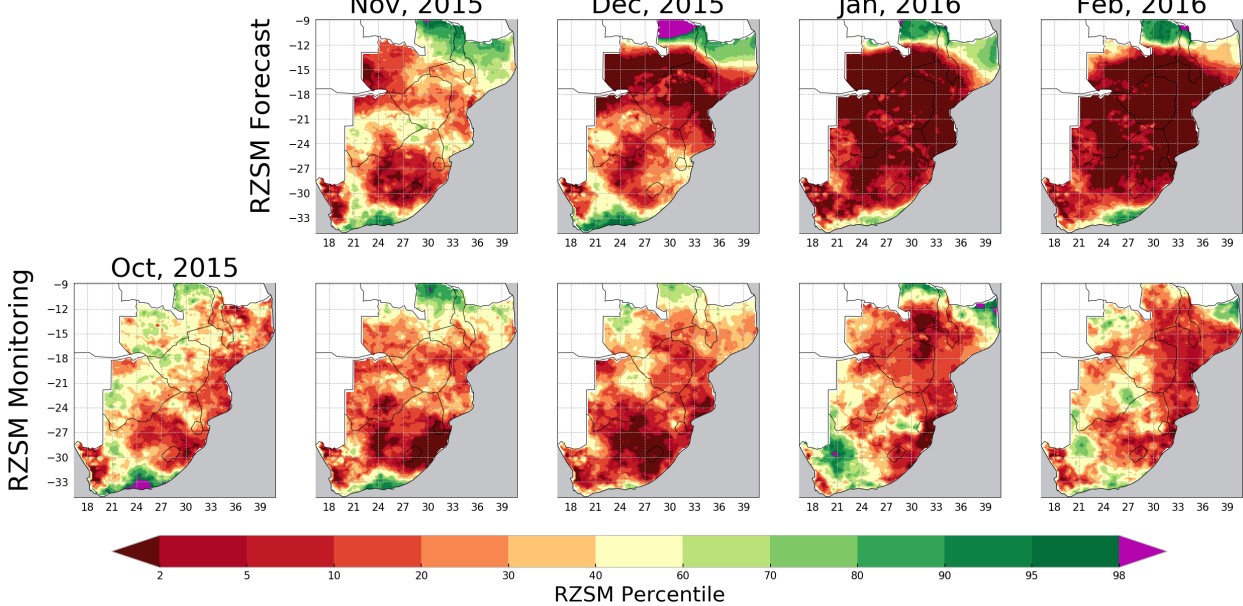


**Figure 4: Forecast (top panel) and Monitoring of Rootzone soil moisture  (RZSM)**

**percentiles for the months of November 2015 through February 2016. October 2015**

**conditions reflect the state of RZSM during the month preceding the forecast**

**initialization on November 1, 2015. The RZSM monitoring product for a given month**

**is available during the early part of the following month. The historical climatology**

**(1982-2010) was used to calculate percentile.**

752

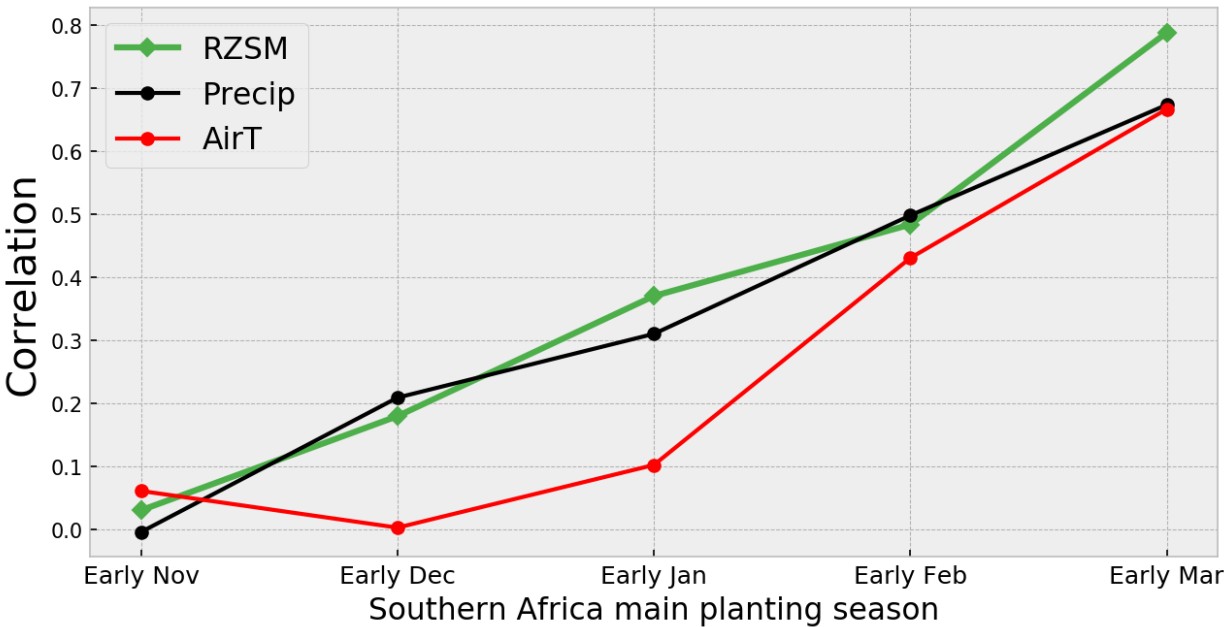

753

**Figure 5: Variability of the correlation between the 3-month seasonal precipitation,**

**3-month seasonal air temperature (AirT), and monthly RZSM monitoring product**

**with the detrended crop yield. This result highlights that RZSM is potentially a better**

**predictor of crop yield than seasonal precipitation and AirT; also, the skill is the**

**highest in early March when DJF seasonal precipitation, AirT, and February RZSM**

**monitoring products are available.**

760

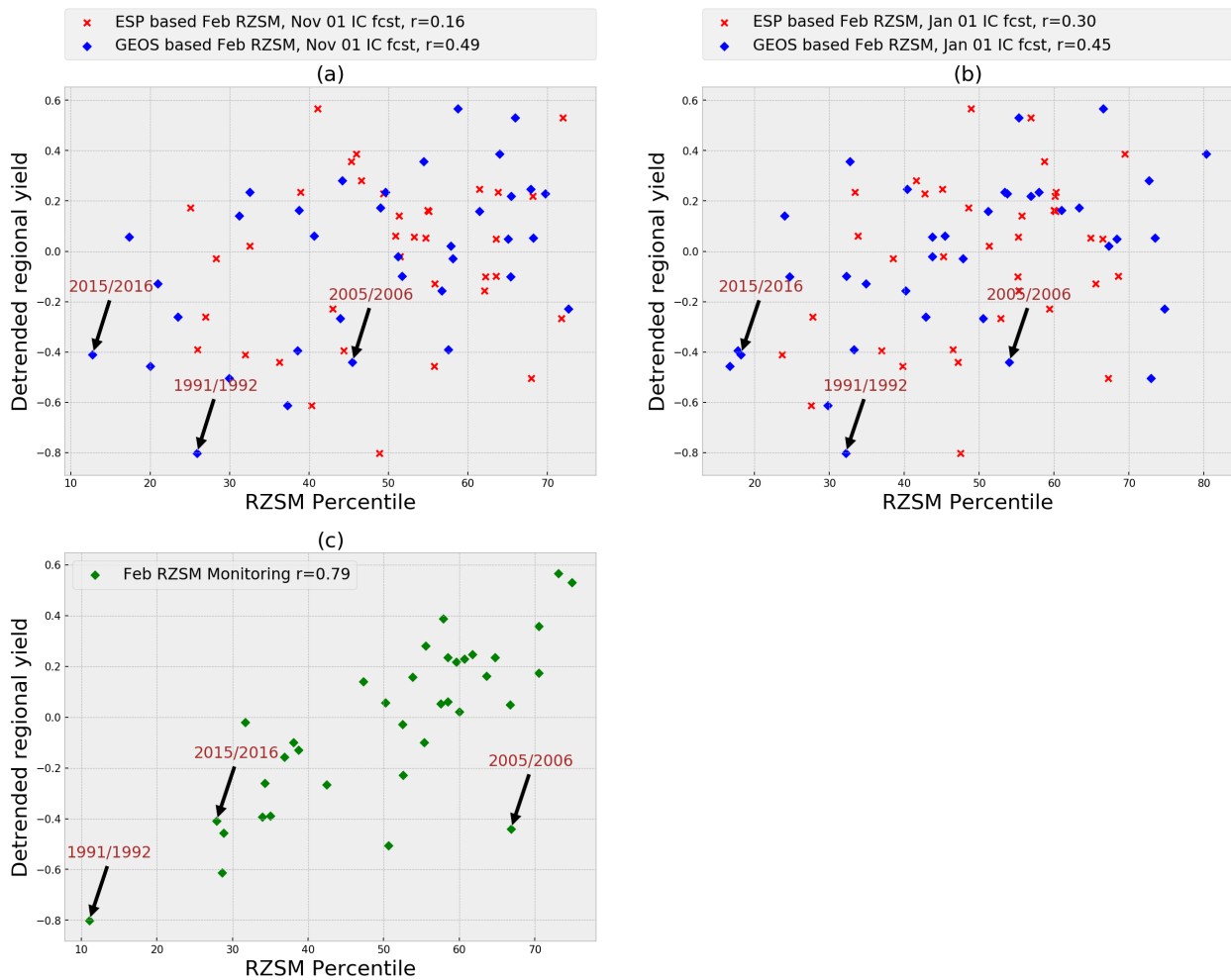

**Figure 6: Covariability of detrended regional yield in southern Africa with: (a) February RZSM forecasts (initialized on November 1) generated using ESP method and bias-corrected GEOS forecasts, (b) February RZSM forecasts (initialized on January 1) generated using ESP method and bias-corrected GEOS forecasts, and (c) the February RZSM monitoring product (available in early March).**

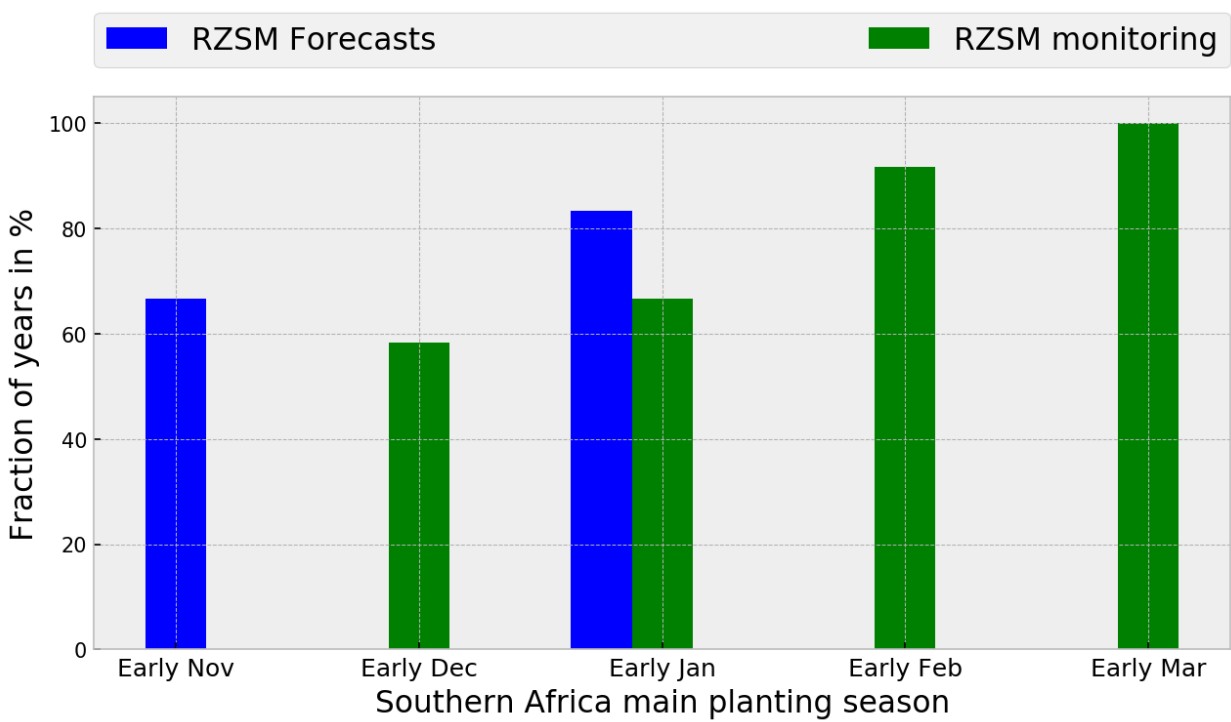



**Figure 7: Fraction of years with below-normal regional crop yield (based on the rank of**
**detrended crop yield) given that the corresponding RZSM forecasts (initialized on**
**November 1 and January 1) and RZSM monitoring product (available in early March)**
**were in the lowest tercile (based on the rank of the RZSM climatology). Note that the Nov 1**
**[Jan 1] RZSM forecasts-based probability of ~66% [~83%] is statistically significant at the**
**~86% [~95%] confidence level.**

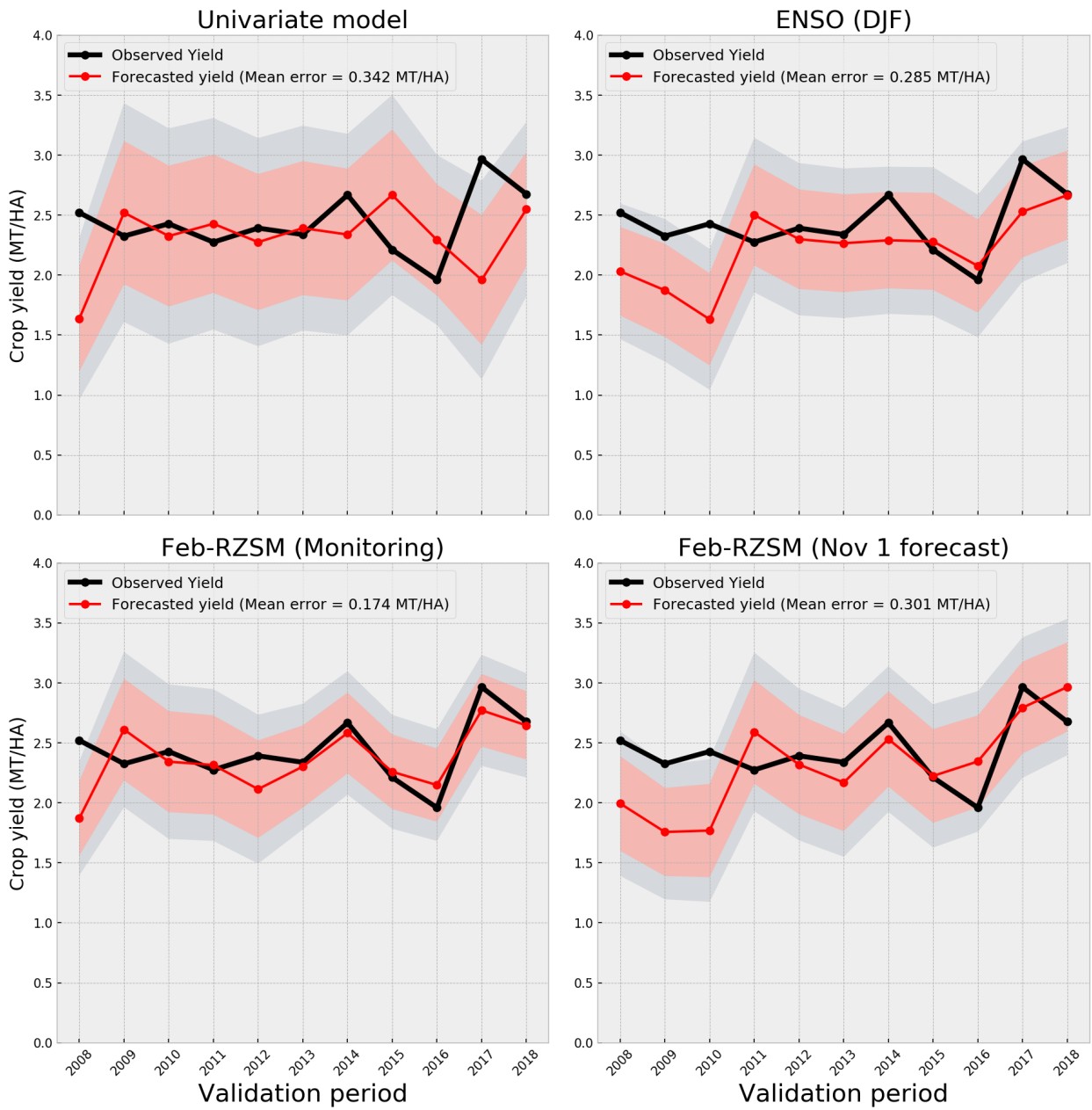

**Figure 8: Comparison of the performance of a Univariate model alone, ENSO (DJF), Feb-RZSM monitoring product, Feb-RZSM forecasting product as a predictor in forecasting crop yield of Southern Africa. Pink [gray] shading indicates 80% [95%] confidence interval.**