# Peer review of "Improving early warning of drought-driven food insecurity in Southern Africa using operational hydrological monitoring and forecasting products"

_Natural Hazards and Earth System Sciences, 2019_

## Referee Comment (RC1) · Anonymous Referee #1 · 19 Sep 2019

To my understanding, this paper connects dynamical forecasts of soil moisture with regional crop yields over southern Africa statistically, and the results are encouraging since the soil moisture forecast correlates with crop yield quite well with a lead time of a few months. This study is novel and has a solid basis on climate-hydrology forecasting, where NASA Hydrological Forecasting and Analysis System that incorporates seasonal climate prediction and land surface hydrological simulation is implemented over southern Africa, and is evaluated for a number of extreme drought cases including the 2015/16 drought during the super El Nino. Utilizing dynamical hydrological forecasts in

agricultural and water resources management sectors is not trivial, and this study push it a step further by smartly combining dynamical and statistical approaches, which provides implications for applications over other regions around the world. The paper is well written and the results are convincing, so I could not comment more while listing only a few minor suggestions below.

1. The abstract could be condensed and reconstructed by placing this southern Africa study in a wider context, where I believe the system has potential to be implemented globally.

2. Two references regarding the African ensemble drought forecasting (Yuan et al., 2013) and southern Africa 2015/16 severe drought attribution (Yuan et al., 2018) might be relevant. The latter focus on rapid evolving soil moisture drought (i.e., flash drought) over southern Africa, where the anthropogenic climate change intensified southern Africa flash drought, especially during 2016/16 El Nino in the midst of heat waves. So, an effective early warning system is essential for drought mitigation over the region.

3. L208-209. The authors mentioned that existing systems like FEWS NET and SADC failed to forecast rainfall during 2015/16. I am wondering whether they can compare the latest GEOS5 rainfall prediction, which is a central component in the forecast system proposed in this study, with those predictions from existing systems. This might highlight the advantage of the new system/method.

4. Although Figure 4 shows a good relationship between crop yield and predicted soil moisture, it might be useful to use some statistical techniques to convert soil moisture prediction into crop yield prediction. Perhaps the authors could comment on that in the discussion section, if they believe it would be useful for their future development of the forecast system.

References:

Yuan, X., L. Wang, and E. F. Wood, 2018: Anthropogenic intensification of southern

African flash droughts as exemplified by the 2015/16 season. Bulletin of the American Meteorological Society, 99, S86-S90, doi:10.1175/BAMS-D-17-007.1

Yuan, X., E. F. Wood, N. W. Chaney, J. Sheffield, J. Kam, M. Liang, and K. Guan, 2013: Probabilistic Seasonal Forecasting of African Drought by Dynamical Models. Journal of Hydrometeorology, 14, 1706-1720, doi:10.1175/JHM-D-13-054.1

---

## Referee Comment (RC2) · Anonymous Referee #2 · 26 Sep 2019

This paper applies a recently developed hydrological forecasting and monitoring system (NHyFAS) to drought early-warning in Southern Africa. This forces a large-scale hydrological model whose parameters depend on global datasets, with 1) observation data and 2) a multi-ensemble forecast. These forcings input in the hydrological model provide monitoring and forecasting hydrological metrics that are then correlated with crop yields to assess their performance as early-warning signals of drought in Southern Africa. Rootzone soil moisture (RZSM) is used as the main hydrological variable for both monitoring and forecasting. With harvest starting in March, authors use monitoring variables available in early Dec, Jan, Feb, Mar (i.e. up to 3-4 months in advance) and monitoring in early Nov and Jan (i.e. up to 4-5 months in advance). Authors test the efficiency of these RZSM products, first on the 2015-16 drought event (with dramatic repercussions on the prices of staple foods) and then on the whole 1982-2018 period (36 years). They show that the proposed forecasting products could have forecast the food availability crisis in Southern Africa in 2015-16 up to 4-5 months before the next harvest starts. They then go on to show that if products are in the lower tercile, there is a high confidence that crop yields will be below average months in advance. Their conclusion is that the proposed products will improve early warning systems of low water-food availability.

The paper's results are interesting, very relevant to this journal and timely, at a time when such early-warning systems for drought conditions are viewed as a priority in Africa (see Nature https://www.nature.com/articles/d41586-019-02760-9). Yet, the text is marred by unstated assumptions, the lack of comparison with existing early-warning systems, and the absence of rationale to explain the results' performance. In particular:

1) The work provides evidence that the proposed products correlate with crop yields, but as the authors know, correlation is not causation. Authors should discuss evidence in the literature of what key variables the forecasts pick up (ENSO maybe?), or alternatively, what supplementary work is needed to establish causation, and therefore, credibility for the products their propose. 2) Other forecasting systems for the area are evoked (Sheffield et al 2014, the African Flood and Drought Monitor (lines 103-104)), why not compare results with those obtained with other products? A justification should be provided in the introduction. 3) If forecasting systems are unavailable, authors link food security crises with El Niño. So that's a simple, well-established indicator (the ENSO index) whose predictive power could easily be compared with that of the RZSM-based products. 4) This journal is an interdisciplinary forum around natural hazards such as droughts and not a hydroclimatology outlet, so authors should make their methods more accessible. A figure of the workflow could help, and so could extra ex-

planations along some of the acronyms. 5) Likewise, justifications for the selection of the key variable (RZSM) or of the forecast ensemble, among others, should be provided to help the paper to be understandable by a larger audience.

I would advise a careful, rigorous revision accounting for the remarks above and where at the minimum, the products' performance should be compared with that of ENSO. If the products work mainly because the forecast ensemble picks up the state of the ENSO index, is there added value to that work:

There is no mention of model/ code / processed data availability for this study: all data sources are the raw data that was used into NHyFAS.

Some detailed comments:

Abstract: it should be made clear in there that the RZSM products are derived from the new NHyFAS. It reads like that they are not.

lines 39-43: authors aren't obligated to show a graph (also that is helpful) but they should cite references.

Line 137: the choice of RZSM as a hydrologic variable of interest makes sense but a rationale should be provided for it being the main (or indeed only) variable of interest in this study. What justifies not using other variables.

Figure 3, and commentary lines 230-250: this seems needlessly confusing. My understanding is that Fig 3 shows the correlation of crop yield with three monitoring-based products whereas the text touts the superiority of the forecasting-based product on all three as early as November. The latter, as well as one of the three monitoring products, includes RZSM, and the distinction is not always clear on first read. Besides, back-and-forth with Figure 4 doesn't make the reading easy either. Could it be a good idea to 1) include the forecasting-based product on Figure 3 to provide a striking visual of why the proposed product is better, and 2) separate comment on Figure 3 from that of Figure 4.

lines 246-249: this should be clarified and explained in Section 2.

Line 266: why the lower tercile? Please justify.

---

## Author Comment (AC1) · 26 Nov 2019

**Reviewer #1**

To my understanding, this paper connects dynamical forecasts of soil moisture with regional crop yields over southern Africa statistically, and the results are encouraging since the soil moisture forecast correlates with crop yield quite well with a lead time of a few months. This study is novel and has a solid basis on climate-hydrology forecasting, where NASA Hydrological Forecasting and Analysis System that incorporates seasonal climate prediction and land surface hydrological simulation is implemented over southern Africa, and is evaluated for a number of extreme drought cases including the 2015/16 drought during the super El Nino. Utilizing dynamical hydrological forecasts in agricultural and water resources management sectors is not trivial, and this study push it a step further by smartly combining dynamical and statistical approaches, which provides implications for applications over other regions around the world. The paper is well written and the results are convincing, so I could not comment more while listing only a few minor suggestions below.

**Response:** Thank you so much for your review and encouraging feedback. Your summary of the study is indeed accurate. Below we have responded to each of your comments. We believe that the revisions made in response to your comments will improve the manuscript.

**Comment #1:**

The abstract could be condensed and reconstructed by placing this southern Africa study in a wider context, where I believe the system has potential to be implemented Globally.

**Response:** Great suggestion. We have revised the abstract to provide a wider context.

**Comment #2:**

Two references regarding the African ensemble drought forecasting (Yuan et al., 2013) and southern Africa 2015/16 severe drought attribution (Yuan et al., 2018) might be relevant. The latter focus on rapid evolving soil moisture drought (i.e., flash drought) over southern Africa, where the anthropogenic climate change intensified southern Africa flash drought, especially during 2016/16 El Nino in the midst of heat waves. So, an effective early warning system is essential for drought mitigation over the region.

**References:**

Yuan, X., L. Wang, and E. F. Wood, 2018: Anthropogenic intensification of southern African flash droughts as exemplified by the 2015/16 season. Bulletin of the American Meteorological Society, 99, S86-S90, doi:10.1175/BAMS-D-17-007.1

Yuan, X., E. F. Wood, N. W. Chaney, J. Sheffield, J. Kam, M. Liang, and K. Guan, 2013: Probabilistic Seasonal Forecasting of African Drought by Dynamical Models. Journal of Hydrometeorology, 14, 1706-1720, doi:10.1175/JHM-D-13-054.1 **Response:** Thank you for suggesting these articles. Both of them have been cited now.

**Comment #3:**

L208-209. The authors mentioned that existing systems like FEWS NET and SADC failed to forecast rainfall during 2015/16. I am wondering whether they can compare the latest GEOS5 rainfall prediction, which is a central component in the forecast system proposed in this study, with those predictions from existing systems. This might highlight the advantage of the new system/method.

**Response:**

Regarding the first sentence, please allow us to clarify that we did not imply that FEWS NET and SADC had failed to forecast rainfall during 2015/16. Please see the corresponding text from the manuscript:

"By this time in the season, both FEWS NET and SADC had provided early warning of poor rainfall performance in the region (Magadzire et al. 2017). The NHyFAS RZSM forecasts would have provided further evidence of a looming unprecedented drought in the region"

FEWS NET and SADC were indicating below normal rainfall in the region during the start of this season, as it was accompanied by one of the strongest El Nino events ever recorded. We argue that RZSM forecasts available on November 1, 2015, (if this system was live back then) would have further substantiated their assessments and actually would have further supported their concerns, as the Nov 1 RZSM forecasts indicated that the RSA, which is one of the main producers in the region, was going to experience the strongest level of drought severity.

The reviewer's point about comparing FEWS NET's and SADC's forecasts with the forecasts from the NHyFAS is fair suggestion. However a direct comparison between NHyFAS forecasts and FEWS NET and SADC forecasts is either not feasible or out of the scope of this study due to the following reasons.

(1) FEWS NET's official forecast is an outlook of food insecurity conditions (see: https://fews.net/) which is based on not just on agroclimatology (i.e., agriculture and climate conditions) but Market conditions, Nutrition and Livelihood conditions. The NHyFAS forecasts which are now being used by FEWS NET would fall into the category of agroclimatological conditions. In fact the goal of the evaluation of the NHyFAS forecasts is to establish if NHyFAS forecasts can be suitable agroclimatological forecast input for FEWS NET to guide the development of food insecurity outlook assessments. Also, FEWS NET Food Insecurity Outlook is based on subjective assessments (in some ways similar to the US drought monitor or US Seasonal Drought Outlook) in addition to quantitative assessments such as agroclimatological forecasts. Finally, FEWS NET's archive of Food Insecurity Outlooks currently spans back to mid-2011 only.

(2) SADC CSC forecasts -- which are typically probabilistic seasonal scale rainfall forecasts, are also based on multiple models (both statistical and dynamical models) as well as expert assessments (hence not entirely based on quantitative inputs). For example, see the text describing the SADC CSC's forecasts methodology behind their latest forecasts below. (Source:

http://csc.sadc.int/en/news-and-events/265-climate-outlook-oct-2019-to-mar-2020)

**" METHODOLOGY**

Using statistical analysis, other climate prediction schemes and expert interpretation, the climate scientists determined likelihoods of above-normal, normal and below-normal rainfall for each area (Figures 1 to 4) for overlapping three-monthly periods i.e. October-November-December (OND), November-December-January (NDJ); December-January-February (DJF); and January- February-March (JFM). Above-normal rainfall is defined as rainfall lying within the wettest third of recorded (30 years, that is, 1971 -2000 mean) rainfall amounts; below-normal is defined as within the driest third of rainfall amounts and normal is the middle third, centred on the climatological median. Figure 5 (a), 5(b), 5(c) and 5(d) show the Long-term (1971-2000) mean rainfall October-November-December, November-December-January, December-January-February and January-February-March season over SADC countries.

The climate scientists took into account oceanic and atmospheric factors that influence our climate over the SADC region, including the El Niño-Southern Oscillation (ENSO), which is currently in its neutral phase. The ENSO is projected to continue in the neutral phase during the entire forecast period. Additional inputs were considered from other global climate prediction centres namely: the European Centre for Medium Range Weather Forecast (ECMWF), National Oceanic and Atmospheric Administration (NOAA), Beijing Climate Centre (BCC), Météo-France, Australian Bureau of Meteorology (BoM), Famine Early Warning Systems Network (FEWSNET), International Research Institute for Climate and Society (IRI), Korea Meteorological Agency, Japan Meteorological Agency (JMA), National Centre for Atmospheric Research (NCAR) and UK Met Office"

Additionally the archive of the purely quantitative forecasts from SADC CSC only goes back to 2017, as can be seen here: http://csc.sadc.int/en/long-range-forecasts

**Comment #4:**

Although Figure 4 shows a good relationship between crop yield and predicted soil moisture, it might be useful to use some statistical techniques to convert soil moisture prediction into crop yield prediction. Perhaps the authors could comment on that in the discussion section, if they believe it would be useful for their future development of the

forecast system.

**Response:** Great point. In response to this comment as well as comments by the Reviewer #2, we have now included an additional analysis in the manuscript, which focuses on predicting regional crop yield using ENSO and RZSM monitoring and forecasting product. Please see the figure below which shows how well the DJF ENSO signal, February RZSM monitoring and forecasting products can forecast regional crop yield in Southern Africa. Crop yield forecasts were made using an AutoRegressive Integrated Moving Average (ARIMA) model. Detailed methodology is included in the manuscript.

Figure: Comparison of observed crop yield with forecasted crop yield estimates made using (from top left corner to the bottom right corner) Autoregression alone, Autoregression and DJF ENSO, Autoregression and Feb RZSM Monitoring product, Autoregression and Feb RZSM forecasting product. Gray shading indicates 95% confidence interval and pink shading indicates 80% confidence interval.

The results indicate that mean error of forecasted yield over 2007-2018 is smallest (0.179 MT/HA) when Feb RZSM (monitoring) product is used as a predictor. When DJF ENSO is used as a predictor the mean error increases to 0.265 MT/HA. Finally when Feb RZSM forecasts, made on Nov 1, are used as a predictor the error level is comparable to DJF ENSO (0.277 MT/HA) but of worth noting is that RZSM forecast based estimates of crop yield are made available 4 months before the crop yield estimates based on DJF ENSO or RZSM (monitoring product). The results also show that adding environmental predictor (RZSM or ENSO) does improve the skill of crop yield forecasts beyond what an autoregressive model alone can provide.

These results highlight the potential value of using RZSM forecasts as well as RZSM monitoring product for predicting regional crop yield in Southern Africa.

---

## Author Comment (AC2) · 26 Nov 2019

**Reviewer #2**

This paper applies a recently developed hydrological forecasting and monitoring system (NHyFAS) to drought early-warning in Southern Africa. This forces a large-scale hydrological model whose parameters depend on global datasets, with 1) observation data and 2) a multi-ensemble forecast. These forcings input in the hydrological model provide monitoring and forecasting hydrological metrics that are then correlated with crop yields to assess their performance as early-warning signals of drought in Southern Africa. Rootzone soil moisture (RZSM) is used as the main hydrological variable for both monitoring and forecasting. With harvest starting in March, authors use monitoring variables available in early Dec, Jan, Feb, Mar (i.e. up to 3-4 months in advance) and monitoring in early Nov and Jan (i.e. up to 4-5 months in advance). Authors test the efficiency of these RZSM products, first on the 2015-16 drought event (with dramatic repercussions on the prices of staple foods) and then on the whole 1982-2018 period (36 years). They show that the proposed forecasting products could have forecast the food availability crisis in Southern Africa in 2015-16 up to 4-5 months before the next harvest starts. They then go on to show that if products are in the lower tercile, there is a high confidence that crop yields will be below average months in advance. Their conclusion is that the proposed products will improve early warning systems of low water-food availability. The paper's results are interesting, very relevant to this journal and timely, at a time when such early-warning systems for drought conditions are viewed as a priority in Africa (see Nature https://www.nature.com/articles/d41586-019-02760-9). Yet, the text is marred by unstated assumptions, the lack of comparison with existing early-warning systems, and the absence of rationale to explain the results' performance.

**Response:** Thank you very much for the review and your constructive comments. Thank you also for pointing us to the Nature article, which we have now cited in the manuscript. Please see our response to your comments below. We hope that these revisions made to respond to your comments would further clarify and substantiate the results. In addition, we have reviewed the method and results sections again to clarify further our methods as/when needed for improved comprehensibility of the manuscript.

**Comment #1**
In particular: 1) The work provides evidence that the proposed products correlate with crop yields, but as the authors know, correlation is not causation. Authors should discuss evidence in the literature of what key variables the forecasts pick up (ENSO maybe?), or alternatively, what supplementary work is needed to establish causation, and therefore, credibility for the products their propose.

**Response:** Thank you for this comment. It is indeed valid. In response to your comment and comment from the reviewer #1, we have now included an additional analysis in the manuscript, which focuses on predicting regional crop yield using ENSO and Rootzone soil moisture monitoring and forecasting product.

Please see the figure below which shows how well the DJF ENSO signal, February RZSM monitoring and forecasting products can forecast regional crop yield in Southern Africa. Crop yield forecasts were made using an AutoRegressive Integrated Moving Average (ARIMA) model. Detailed methodology is included in the manuscript.

[Figure]

*Figure: Comparison of observed crop yield with forecasted crop yield estimates made using (from top left corner to the bottom right corner) Autoregression alone, Autoregression and DJF ENSO, Autoregression and Feb RZSM Monitoring product, Autoregression and Feb RZSM forecasting product. Gray shading indicates 95% confidence interval and pink shading indicates 80% confidence interval.*

The results indicate that mean error of forecasted yield over 2007-2018 is smallest (0.179 MT/HA) when Feb RZSM (monitoring) product is used as a predictor. When DJF ENSO is used as a predictor the mean error increases to 0.265 MT/HA. Finally when Feb RZSM forecasts, made on Nov 1, are used as a predictor the error level is comparable to DJF ENSO (0.277 MT/HA) but of worth noting is that RZSM forecast based estimates of crop yield are made available 4 months before the crop yield estimates based on DJF ENSO or RZSM (monitoring product). The results also show that adding environmental predictor (RZSM or ENSO) does improve the skill of crop yield forecasts beyond what an autoregressive model alone can provide.

These results highlight the potential value of using RZSM forecasts as well as RZSM monitoring product for predicting regional crop yield in Southern Africa.

**Comment #2 and #3:**
Other forecasting systems for the area are evoked (Sheffield et al 2014, the African Flood and Drought Monitor (lines 103-104)), why not compare results with those obtained with other products? A justification should be provided in the introduction. If forecasting systems are unavailable, authors link food security crises with El Niño. So that's a simple, well-established indicator (the ENSO index) whose predictive power could easily be compared with that of the RZSM-based products.

**Response:** Typically the forecast outputs from seasonal forecasting systems are only available in the form of images through web-portals and more importantly historical forecasts are not available for a sufficiently long enough period. For example (as accessed on November 22nd, 2019). the Africa Flood and Drought monitor only provides access to seasonal forecasts from April 2018 to September 2018.

It is also worth noting that the NHyFAS is the only seasonal hydrologic forecasting system for Africa that is based on the The Climate Hazards Infrared Precipitation with Stations (CHIRPS) (https://www.nature.com/articles/sdata201566) which benefits from the satellite era precipitation estimates as well as greater access to stations based precipitation measurements, hence over Africa, is of higher quality. Several past studies have indicated that too. Reliance on high quality precipitation dataset allows for an improved climatology of simulated hydrologic variables (such as RZSM) and improvement in hydrologic initial conditions (which are a substantial source of skill in any seasonal scale hydrologic forecasting system).

We have now added the above discussion in the manuscript.  We have also addressed your comment on comparison with ENSO by including a new analysis. Please see our response to comment #1.

**Comment #4**
This journal is an interdisciplinary forum around natural hazards such as droughts and not a hydroclimatology outlet, so authors should make their methods more accessible. A figure of the workflow could help, and so could extra explanations along some of the acronyms.

**Response:** Makes sense. Initially we had not provided details on the setup of the NHyFAS as it is described in Arsenault et al. (in review), which is the key paper on this system. However, now we have added the following flow chart in the manuscript. This flow chart provides an overview of the process to get gridded RZSM percentile and also defines the hydrologic forecast-related acronyms.

[Figure]

*Figure: Flow diagram of the process and inputs to generate RZSM forecast percentiles*

Reference: Arsenault, K.R., Shukla, S., Hazra, A., Getirana, A., McNally, A., Kumar, S.V., Koster, R.D., Peters-Lidard, C.D., Zaitchik, B.F., Badr, H., Jung, H.C., Narapusetty, B., Navari, M., Wang, S., Mocko, D., Funk, C., Harrison, L., Husak, G.J., Adoum, A., Galu, G., Magadzire, T., Roningen, J., Shaw, M., Eylander, J., Bergaoui, K., McDonnell, R.A., and Verdin, J.P., 2019, The NASA hydrological forecast system for food and water security applications. Bulletin of the American Meteorological Society, In review.water security applications. Bulletin of the American Meteorological Society (in review)

**Comment #5**
Likewise, justifications for the selection of the key variable (RZSM) or of the forecast ensemble, among others, should be provided to help the paper to be understandable by a larger audience.

**Response:** Good point. We have now added the following text in the manuscript.

*Rootzone SM (RZSM) is the main hydrologic variable used in this analysis. RZSM indicates the soil moisture in the top one meter of the soil profile. Typically the length of the root of crops such as maize (main crop in the region of SA) close to one meter, hence the choice of RZSM as the key forecast variables. Moreover the entire depth of the soil profile is different for the two models used in this analysis, typically about 2 m for Noah-MP and about 4 m for CLSM, hence RZSM also allows for a consistent way to merge soil moisture products from both models.*

**Comment #6**
I would advise a careful, rigorous revision accounting for the remarks above and where at the minimum, the products' performance should be compared with that of ENSO. If the products work mainly because the forecast ensemble picks up the state of the ENSO index, is there added value to that work.

**Response:** Please see our response to your comment #2 and #3, which addresses this comment as well.

**Comment #7**
There is no mention of model/ code / processed data availability for this study: all data sources are the raw data that was used into NHyFAS.

**Response:** Good point. We have mentioned in the manuscript where the models source code and input 'observed' forcings data can be found. The maps of output seasonal forecasts are also available for public access. Bias-corrected seasonal forecasts and hydrologic forecasts (e.g., RZSM) data are currently not available for public access. We anticipate though that those forecasts will eventually be available for public access from NASA web-services, similar to other NASA and FEWS NET supported land data assimilation (FLDAS) outputs.

Some detailed comments:

**Comment #8**

Abstract: it should be made clear in there that the RZSM products are derived from the new NHyFAS. It reads like that they are not.

**Response:** Done.

**Comment #9**

lines 39-43: authors aren't obligated to show a graph (also that is helpful) but they should cite references.

**Response:** We have added the following figure showing that the percentage of income held by the bottom 10% and 20% of the population has not changed much in the region.

[Figure]

[Figure]

**Comment #10**

Line 137: the choice of RZSM as a hydrologic variable of interest makes sense but a rationale should be provided for it being the main (or indeed only) variable of interest in this study. What justifies not using other variables.

**Response:** Please see our response to your comment #5.

**Comment #11**

Figure 3, and commentary lines 230-250: this seems needlessly confusing. My understanding is that Fig 3 shows the correlation of crop yield with three monitoring-based products whereas the text touts the superiority of the forecasting-based product on all three as early as November. The latter, as well as one of the three monitoring products, includes RZSM, and the distinction is not always clear on first read.

Besides, back-and-forth with Figure 4 doesn't make the reading easy either. Could it be a good idea to 1) include the forecasting-based product on Figure 3 to provide a striking visual of why the proposed product is better, and 2) separate comment on Figure 3 from that of Figure 4.

**Response:** Sorry about the confusion. We have now revised the texts discussing the results of Figure 3 and 4. We now discuss each of those figures in distinct sub-sections. The goal of the Figure 3 (that shows the correlation between monitoring products such as RZSM, seasonal precipitation and Air temperature with crop yield) is to:

(1) Examine how the correlation changes as the season progresses.
(2) Which variable and when has the strongest correlation occurs with crop yield.

Based on our Figure 3, we identify February RZSM to be the variable with the strongest correlation with the crop yield. Once that is established, Figure 4 then focuses on examining how well the forecast of Feb RZSM made on November 1 and January 1 correlates with crop yield.

**Comment #12**

Line 266: why the lower tercile? Please justify

**Response:** Southern Africa is a mostly rainfed region, hence the crop yield is generally below normal during drought years as evident by several drought years in the recent past (2014-15, 2015-16, 2018-19). Thus in order to evaluate the performance of NHyFAS monitoring and forecasting products in identifying below normal crop yield, we focused on the years when the RZSM monitoring and forecasting products were in the lowest tercile (bottom 12 out of 36 values) as those events represent drought years. We have now added this text in the manuscript as well.

---

## Author Response (AR1)

**Response:**

Dear Reviewers, Thank you so much for your constructive feedback and comments. We have revised the manuscript based on your comments, and we believe that the manuscript is improved due to those changes. Below, we provide point-by-point responses to each of your comments. We have also included any new texts/figures added to manuscript in response to each comment. New texts have been italicized and highlighted in light yellow color.

**Reviewer #1**

To my understanding, this paper connects dynamical forecasts of soil moisture with regional crop yields over southern Africa statistically, and the results are encouraging since the soil moisture forecast correlates with crop yield quite well with a lead time of a few months. This study is novel and has a solid basis on climate-hydrology forecasting, where NASA Hydrological Forecasting and Analysis System that incorporates seasonal climate prediction and land surface hydrological simulation is implemented over southern Africa, and is evaluated for a number of extreme drought cases including the 2015/16 drought during the super El Nino. Utilizing dynamical hydrological forecasts in agricultural and water resources management sectors is not trivial, and this study push it a step further by smartly combining dynamical and statistical approaches, which provides implications for applications over other regions around the world. The paper is well written and the results are convincing, so I could not comment more while listing only a few minor suggestions below.

**Response:** Thank you so much for your review and encouraging feedback. Your summary of the study is indeed accurate. Below, we have responded to each of your comments. We believe that the revisions made in response to your comments will improve the quality of the manuscript.

**Comment #1:**
The abstract could be condensed and reconstructed by placing this southern Africa study in a wider context, where I believe the system has potential to be implemented Globally.

**Response:** Great suggestion. We have trimmed the abstract and added the following sentence to texts to highlight global implications of this study.

Text added to the abstract:

*Finally, since a framework similar to NHyFAS can be used to provide RZSM monitoring and forecasting products over other regions of the globe, this case study also demonstrates potential for supporting food insecurity early warning globally*

Text added to the Conclusion:

*This study demonstrates the value of the NHyFAS products in supporting food insecurity early warning in the SA region. It is worth mentioning that since NHyFAS currently covers Africa and the Middle East region, the NHyFAS products are applicable for food insecurity early warning in the rest of Africa and the Middle East as well. Based on this study, it is postulated (future research pending) that NHyFAS RZSM products can be particularly effective for those rainfed agriculture regions and seasons which are not known to have strong teleconnection (e.g. with ENSO), as in the SA region. Finally, since the data sets and models used to impelement the NHyFAS are available globally, a similar seasonal RZSM monitoring and forecasting framework can be developed at a global scale to support food insecurity early warning in other rainfed regions across the globe.*

**Comment #2:**
Two references regarding the African ensemble drought forecasting (Yuan et al., 2013) and southern Africa 2015/16 severe drought attribution (Yuan et al., 2018) might be relevant. The latter focus on rapid evolving soil moisture drought (i.e., flash drought) over southern Africa, where the anthropogenic climate change intensified southern Africa flash drought, especially during 2016/16 El Nino in the midst of heat waves. So, an effective early warning system is essential for drought mitigation over the region.

References:
Yuan, X., L. Wang, and E. F. Wood, 2018: Anthropogenic intensification of southern African flash droughts as exemplified by the 2015/16 season. Bulletin of the American Meteorological Society, 99, S86-S90, doi:10.1175/BAMS-D-17-007.1

Yuan, X., E. F. Wood, N. W. Chaney, J. Sheffield, J. Kam, M. Liang, and K. Guan, 2013: Probabilistic Seasonal Forecasting of African Drought by Dynamical Models. Journal of Hydrometeorology, 14, 1706-1720, doi:10.1175/JHM-D-13-054.1

**Response:** Thank you for suggesting these articles. Both of them have now been cited.

**Comment #3:**
L208-209. The authors mentioned that existing systems like FEWS NET and SADC failed to forecast rainfall during 2015/16. I am wondering whether they can compare the latest GEOS5 rainfall prediction, which is a central component in the forecast system proposed in this study, with those predictions from existing systems. This might highlight the advantage of the new system/method.

**Response:**

Regarding the first sentence, please allow us to clarify that we did not imply that FEWS NET and SADC had failed to forecast rainfall during 2015/16. Please see the corresponding text from the manuscript:

"By this time in the season, both FEWS NET and SADC had provided early warning of poor rainfall performance in the region (Magadzire et al. 2017). The NHyFAS RZSM forecasts would have provided further evidence of a looming unprecedented drought in the region"

FEWS NET and SADC were indicating below-normal rainfall in the region during the start of this season, as it was accompanied by one of the strongest El Niño events ever recorded. We argue that RZSM forecasts available on November 1, 2015, (if this system were live back then) would have further substantiated their assessments and actually further supported their concerns, as the Nov. 1 RZSM forecasts indicated that the RSA, which is one of the main producers in the region, was going to experience the strongest level of drought severity.

The reviewer's point about comparing FEWS NET and SADC's forecasts with the forecasts from the NHyFAS is fair suggestion. However, a direct comparison between NHyFAS forecasts and FEWS NET and SADC forecasts is not feasible. We have included the following section in the manuscript, in response:

*4.1      Comparison with existing drought forecasting systems and approaches:*

*In this study, we keep the comparison with existing forecasting systems and approaches*

*limited to the comparison of the performance of NHyFAS products with (i) ESP (i.e. climatology)*

*based RZSM forecasts and (ii) ENSO-based crop yield forecasts, both of which are commonly used approaches for drought forecasting in the region, including by early warning agencies such as FEWS NET. Comparison against both approaches shows clear added value of using the NHyFAS products. We could not compare the performance of the NHyFAS with FEWS NET or SADC's official historical forecasts because:*

*(i) FEWS NET's official forecast is an outlook of food insecurity conditions (Funk et al. 2019) (https://fews.net/) which is based not only on agroclimatology (i.e., agriculture and climate conditions) but also on market conditions and nutrition and livelihood conditions. The NHyFAS forecasts that are now being used by FEWS NET would fall into the category of agroclimatological conditions. In fact, the goal of the evaluation of the NHyFAS forecasts is to establish whether NHyFAS forecasts can be suitable agroclimatological forecast inputs for FEWS NET to guide the development of food insecurity outlook assessments. Also, FEWS NET Food Insecurity Outlook is partly based on subjective assessments, in some ways similar to the U.S. drought monitor (Svoboda et al., 2002) or U.S. Seasonal Drought Outlook, in addition to quantitative assessments such as agroclimatological forecasts. Finally, FEWS NET's archive of Food Insecurity Outlooks currently extends back only to mid-2011.*

*(ii) SADC CSC's issues probabilistic seasonal-scale rainfall forecasts. These forecasts are based on multiple models (both statistical and dynamical) as well as subjective expert assessments, which makes comparison with purely quantitative products inappropriate. Additionally, the archive of purely quantitative forecasts from SADC CSC only goes back to 2017.*

*Finally, the NHyFAS products are intended to be used as an addition to the existing early warning tools of FEWS NET and SADC CSC, which are partners in the efforts described in this study, rather than replacing any of the existing tools."*

**Comment #4:**

Although Figure 4 shows a good relationship between crop yield and predicted soil moisture, it might be useful to use some statistical techniques to convert soil moisture prediction into crop yield prediction. Perhaps the authors could comment on that in the discussion section, if they believe it would be useful for their future development of the forecast system.

**Response:** Great point. In response to this comment, as well as comments by the Reviewer #2, we have now included an additional analysis in the following section. Shown below (and in the manuscript) are Figure 8 and Table 1, which summarize the results of this section.

**3.3 *Performance of NHyFAS in providing routine operational crop yield forecasts**

*Finally, we evaluate the performance of NHyFAS for supporting food insecurity early warning in SA by examining the accuracy of RZSM monitoring and RZSM forecasting products in predicting regional crop yields. We compare the crop yield forecasts made with the RSZM products against both univariate forecasts (using only past observed crop yields) and forecasts made with ENSO. As ENSO is a widely used predictor for precipitation and crop yield forecasts in this region, we examine the added value of using NHyFAS RZSM monitoring and forecasting products above and beyond ENSO. All forecasts are done using ARIMA models described in section 2.6.*

*Figure 8 shows a comparison between the "observed" reported crop yield (black lines) and the "out-of-sample" (i.e. post-training period) forecasted yield (red lines) produced with a univariate model, and the models using environmental exogenous predictors (i) DJF ENSO, (ii) Feb-RZSM (monitoring) product, (iii) Feb-RZSM (Forecasting product) initialized on Nov. 1., in addition to that univariate model.*

*The results indicate that: (i) environmental predictors such as ENSO and the NHyFAS products can make crop yield forecasts that are more accurate than those produced using only a univariate approach. When ENSO is used as an additional predictor (in addition to a Univariate*

model), the MAE reduces from 0.342 MT/HA to 0.285 MT/HA, a ~17% reduction in error. (ii) Use of the Feb-RZSM monitoring product has an even larger impact, reducing the MAE by about 50%, to 0.174 MT/HA. (iii) Use of the Feb-RZSM forecasting product (initialized on Nov 1) has an impact similar to that of DJF ENSO. Although the MAE is about 6% larger when the forecasting product is used rather than the ENSO predictor, the forecasting product has the significant advantage of being available for about 4 months earlier. For comparison (not shown here) MAE of Feb-RZSM forecasting product (initialized on Nov 1) is slightly smaller (~6%) than the MAE of August-October (ASO)-ENSO (also available in early Nov) and is comparable to the MAE of September-November (SON)-ENSO (available in early December) as a predictor of crop yield forecast.

Table 1 shows the number of times the observed yield is within the 80% confidence interval of the forecasts and the mean spread of the confidence interval. The improvement in performance obtained when the Feb-RZSM monitoring product is used is clear; during 10 of the 11 years in the validation period, the observed yield falls within the 80% confidence interval, whereas this happens in only 7 years when DJF ENSO is used as the additional predictor. The mean spread of the confidence interval associated with the use of the Feb-RZSM monitoring product (0.70 MT/HA) is also the smallest.

Table 1: Performance of "out-of-sample" crop yield forecasting over the validation period of 2008-2018.

| | Univariate model | Univariate model + ENSO | Univariate model + Feb-RZSM (Monitoring) | Univariate model + Feb-RZSM (forecast) |
|---|---|---|---|---|
| **Mean absolute error over the validation period (MT/HA)** | 0.342 | 0.285 | 0.174 | 0.301 |
| **Number of years observed yield is within 95% confidence interval bound** | 9 | 10 | 10 | 9 |
| **Mean spread of 95% confidence interval (MT/HA)** | 1.64 | 1.20 | 1.07 | 1.20 |
| **Number of years observed yield is within 80% confidence interval bound** | 9 | 7 | 10 | 7 |
| **Mean spread of 80% confidence interval (MT/HA)** | 1.07 | 0.78 | 0.70 | 0.78 |

[Figure]

**Figure 8: Comparison of the performance of a Univariate model alone, ENSO (DJF), Feb-RZSM monitoring product, Feb-RZSM forecasting product as a predictor in forecasting crop yield of Southern Africa. Pink [gray] shading indicates 80% [95%] confidence interval.**

**Reviewer #2**

This paper applies a recently developed hydrological forecasting and monitoring system (NHyFAS) to drought early-warning in Southern Africa. This forces a large-scale hydrological model whose parameters depend on global datasets, with 1) observation data and 2) a multi-ensemble forecast. These forcings input in the hydrological model provide monitoring and forecasting hydrological metrics that are then correlated with crop yields to assess their performance as early-warning signals of drought in Southern Africa. Rootzone soil moisture (RZSM) is used as the main hydrological variable for both monitoring and forecasting. With harvest starting in March, authors use monitoring variables available in early Dec, Jan, Feb, Mar (i.e. up to 3-4 months in advance) and monitoring in early Nov and Jan (i.e. up to 4-5 months in advance). Authors test the efficiency of these RZSM products, first on the 2015-16 drought event (with dramatic repercussions on the prices of staple foods) and then on the whole 1982-2018 period (36 years). They show that the proposed forecasting products could have forecast the food availability crisis in Southern Africa in 2015-16 up to 4-5 months before the next harvest starts. They then go on to show that if products are in the lower tercile, there is a high confidence that crop yields will be below average months in advance. Their conclusion is that the proposed products will improve early warning systems of low water-food availability. The paper's results are interesting, very relevant to this journal and timely, at a time when such early-warning systems for drought conditions are viewed as a priority in Africa (see Nature https://www.nature.com/articles/d41586-019-02760-9). Yet, the text is marred by unstated assumptions, the lack of comparison with existing early-warning systems, and the absence of rationale to explain the results' performance.

**Response:** Thank you very much for the review and for your constructive comments. Thank you also for pointing us to the *Nature* article, which we have now cited in the manuscript. Please see our response to your comments below. We hope that these revisions made in response to your comments will further clarify and substantiate the results. In addition, we have reviewed the method and results sections again to clarify our methods as/when needed for improved comprehensibility of the manuscript.

**Comment #1**
In particular: 1) The work provides evidence that the proposed products correlate with crop yields, but as the authors know, correlation is not causation. Authors should discuss evidence in the literature of what key variables the forecasts pick up (ENSO maybe?), or alternatively, what supplementary work is needed to establish causation, and therefore, credibility for the products their propose.

**Response:** Thank you for this comment. It is indeed valid. In response to your comment and comment from reviewer #1, we have now included an additional analysis in the manuscript, in the following section. Shown below (and in the manuscript) are Figure 8 and Table 1, which summarize the results of this section.

**3.3    *Performance of NHyFAS in providing routine operational crop yield forecasts**

*Finally, we evaluate the performance of NHyFAS for supporting food insecurity early warning in SA by examining the accuracy of RZSM monitoring and RZSM forecasting products in predicting regional crop yields. We compare the crop yield forecasts made with the RSZM products against both univariate forecasts (using only past observed crop yields) and forecasts made with ENSO. As ENSO is a widely used predictor for precipitation and crop yield forecasts in this region, we examine the added value of using NHyFAS RZSM monitoring and forecasting products above and beyond ENSO. All forecasts are done using ARIMA models described in section 2.6.*

*Figure 8 shows a comparison between the "observed" reported crop yield (black lines) and the "out-of-sample" (i.e. post-training period) forecasted yield (red lines) produced with a univariate model, and the models using environmental exogenous predictors (i) DJF ENSO, (ii) Feb-RZSM (monitoring) product, (iii) Feb-RZSM (Forecasting product) initialized on Nov. 1., in addition to that univariate model.*

*The results indicate that: (i) environmental predictors such as ENSO and the NHyFAS products can make crop yield forecasts that are more accurate than those produced using only a univariate approach. When ENSO is used as an additional predictor (in addition to a Univariate model), the MAE reduces from 0.342 MT/HA to 0.285 MT/HA, a ~17% reduction in error. (ii) Use of the Feb-RZSM monitoring product has an even larger impact, reducing the MAE by about*

*50%, to 0.174 MT/HA. (iii) Use of the Feb-RZSM forecasting product (initialized on Nov 1) has an impact similar to that of DJF ENSO. Although the MAE is about 6% larger when the forecasting product is used rather than the ENSO predictor, the forecasting product has the significant advantage of being available for about 4 months earlier. For comparison (not shown here) MAE of Feb-RZSM forecasting product (initialized on Nov 1) is slightly smaller (~6%) than the MAE of August-October (ASO)-ENSO (also available in early Nov) and is comparable to the MAE of September-November (SON)-ENSO (available in early December) as a predictor of crop yield forecast.*

*Table 1 shows the number of times the observed yield is within the 80% confidence interval of the forecasts and the mean spread of the confidence interval. The improvement in performance obtained when the Feb-RZSM monitoring product is used is clear; during 10 of the 11 years in the validation period, the observed yield falls within the 80% confidence interval, whereas this happens in only 7 years when DJF ENSO is used as the additional predictor. The mean spread of the confidence interval associated with the use of the Feb-RZSM monitoring product (0.70 MT/HA) is also the smallest.*

Table 1: Performance of "out-of-sample" crop yield forecasting over the validation period of 2008-2018.

|  | Univariate model | Univariate model + ENSO | Univariate model + Feb-RZSM (Monitoring) | Univariate model + Feb-RZSM (forecast) |
|---|---|---|---|---|
| **Mean absolute error over the validation period (MT/HA)** | 0.342 | 0.285 | 0.174 | 0.301 |
| **Number of years observed yield is within 95% confidence interval bound** | 9 | 10 | 10 | 9 |
| **Mean spread of 95% confidence interval (MT/HA)** | 1.64 | 1.20 | 1.07 | 1.20 |
| **Number of years observed yield is within 80% confidence interval bound** | 9 | 7 | 10 | 7 |
| **Mean spread of 80% confidence interval (MT/HA)** | 1.07 | 0.78 | 0.70 | 0.78 |

[Figure]

**Figure 8: Comparison of the performance of a Univariate model alone, ENSO (DJF), Feb-RZSM monitoring product, Feb-RZSM forecasting product as a predictor in forecasting crop yield of southern Africa. Pink [gray] shading indicates 80% [95%] confidence interval.**

**Comment #2 and #3:**

Other forecasting systems for the area are evoked (Sheffield et al 2014, the African Flood and Drought Monitor (lines 103-104), why not compare results with those obtained with other products? A justification should be provided in the introduction. If forecasting systems are unavailable, authors link food security crises with El Niño. So that's a simple, well-established indicator (the ENSO index) whose predictive power could easily be compared with that of the RZSM-based products.

**Response:**

Thank you for this important comment. In response to your comment and a comment from reviewer #1, we have added the following two sections to the manuscript.

1. " 3.3    Performance of NHyFAS in providing routine operational crop yield forecasts" This section provides a comparison of crop yield forecast skill based on the NHyFAS products with ENSO. Please also see our response to your comment #1.
2. "4.1    Comparison with existing drought forecasting systems and approaches"

*4.1      Comparison with existing drought forecasting systems and approaches:*

*In this study, we keep the comparison with existing forecasting systems and approaches limited to the comparison of the performance of NHyFAS products with (i) ESP (i.e. climatology) based RZSM forecasts and (ii) ENSO-based crop yield forecasts, both of which are commonly used approaches for drought forecasting in the region, including by early warning agencies such as FEWS NET. Comparison against both approaches shows clear added value of using the NHyFAS products. We could not compare the performance of the NHyFAS with FEWS NET or SADC's official historical forecasts because:*

*(i) FEWS NET's official forecast is an outlook of food insecurity conditions (Funk et al. 2019) (https://fews.net/) which is based not only on agroclimatology (i.e., agriculture and climate conditions) but also on market conditions and nutrition and livelihood conditions. The NHyFAS forecasts that are now being used by FEWS NET would fall into the category of agroclimatological conditions. In fact, the goal of the evaluation of the NHyFAS forecasts is to establish whether NHyFAS forecasts can be suitable agroclimatological forecast inputs for*

*FEWS NET to guide the development of food insecurity outlook assessments. Also, FEWS NET*

*Food Insecurity Outlook is partly based on subjective assessments, in some ways similar to the*

*U.S. drought monitor (Svoboda et al., 2002) or U.S. Seasonal Drought Outlook, in addition to*

*quantitative assessments such as agroclimatological forecasts. Finally, FEWS NET's archive of*

*Food Insecurity Outlooks currently extends back only to mid-2011.*

*(ii) SADC CSC's issues probabilistic seasonal-scale rainfall forecasts. These forecasts are based*

*on multiple models (both statistical and dynamical) as well as subjective expert assessments,*

*which makes comparison with purely quantitative products inappropriate. Additionally, the*

*archive of purely quantitative forecasts from SADC CSC only goes back to 2017.*

*Finally, the NHyFAS products are intended to be used as an addition to the existing early*

*warning tools of FEWS NET and SADC CSC, which are partners in the efforts described in this*

*study, rather than replacing any of the existing tools.*

Finally, regarding the comparison with the Africa Flood and Drought monitor we note that, as accessed on November 22nd, 2019, the Africa Flood and Drought monitor only provides access to seasonal forecasts from April 2018 to September 2018. Additionally, it is also worth noting that the NHyFAS is the only seasonal hydrologic forecasting system for Africa that is based on the The Climate Hazards InfraRed Precipitation with Stations (CHIRPS) (https://www.nature.com/articles/sdata201566), which benefits from the satellite era precipitation estimates, as well as greater access to station based precipitation measurements, hence over Africa, is of higher quality. Several past studies have indicated that too. Reliance on a high-quality precipitation data set allows for an improved climatology of simulated hydrologic variables (such as RZSM) and improvement in hydrologic initial conditions (which are a substantial source of skill in any seasonal scale hydrologic forecasting system).

**Comment #4**
This journal is an interdisciplinary forum around natural hazards such as droughts and not a hydroclimatology outlet, so authors should make their methods more accessible. A figure of the workflow could help, and so could extra explanations along some of the acronyms.

**Response:** We understand the logic behind this comment. Initially, we had not provided details on the setup of the NHyFAS, as it is described in Arsenault et al. (in review), which is the key paper on this system. However, we have now added the following flowchart to the manuscript. This flowchart provides an overview of the process to get gridded RZSM percentile, and also defines the hydrologic forecast-related acronyms.

[Figure]

**Figure 3: Overview of the NHyFAS implementation to produce RZSM monitoring and forecasting products, as used in this study.**

Reference: Arsenault, K.R., Shukla, S., Hazra, A., Getirana, A., McNally, A., Kumar, S.V., Koster, R.D., Peters-Lidard, C.D., Zaitchik, B.F., Badr, H., Jung, H.C., Narapusetty, B., Navari, M., Wang, S., Mocko, D., Funk, C., Harrison, L., Husak, G.J., Adoum, A., Galu, G., Magadzire, T., Roningen, J., Shaw, M., Eylander, J., Bergaoui, K., McDonnell, R.A., and Verdin, J.P., 2019, The NASA hydrological forecast system for food and water security applications. Bulletin of the American Meteorological Society, In review.water security applications. Bulletin of the American Meteorological Society (in review)

**Comment #5**
Likewise, justifications for the selection of the key variable (RZSM) or of the forecast ensemble, among others, should be provided to help the paper to be understandable by a larger audience.

**Response:** Good point. We have now added the following text in the manuscript, in section 2.4:

*The performance of the NHyFAS system is evaluated mainly through its RZSM monitoring (generated from OL) and forecasting products. RZSM indicates the soil moisture in the top one meter of the soil profile. Typically, the length of the roots of crops such as maize (main crop in the region of SA) is close to one meter, hence the choice of RZSM as the key forecast variable. Moreover, the entire depth of the soil profile is different for the two models used in this analysis, typically about 2 m for Noah-MP and about 4 m for CLSM; hence RZSM also allows for a consistent way to merge soil moisture products from both models.*

**Comment #6**

I would advise a careful, rigorous revision accounting for the remarks above and where at the minimum, the products' performance should be compared with that of ENSO. If the products work mainly because the forecast ensemble picks up the state of the ENSO index, is there added value to that work.

**Response:** Please see our response to your comments #2 and #3, which addresses this comment as well.

**Comment #7**

There is no mention of model/ code / processed data availability for this study: all data sources are the raw data that was used into NHyFAS.

**Response:** Good point. We have included the following text in the acknowledgements:

*GEOS forecast data sets are generated and supported by NASA's Global Modeling and Assimilation Office (GMAO). Model source code can be found at NASA's Land Information System's GitHub repository (https://lis.gsfc.nasa.gov/news/latest-lis-code-now-available-github). Model parameters are available through email request. The daily CHIRPS precipitation data can be found here (ftp://ftp.chg.ucsb.edu/pub/org/chg/products/CHIRPS-2.0/global_daily/netcdf/p25/). MERRA-2 reanalysis-based atmospheric forcings can be found through NASA's GES DISC archive (https://disc.gsfc.nasa.gov/datasets?keywords=%22MERRA-2%22&page=1&source=Models%2FAnalyses%20MERRA-2). NHyFAS forecasts, in the form of maps, can be found here https://lis.gsfc.nasa.gov/projects/nhyfas. As of now, NHyFAS forecast*

*data sets are not publicly accessiblePlease note that bias-corrected seasonal forecasts and hydrologic forecasts (e.g., RZSM) data are currently not available for public access. We anticipate though that those forecasts will eventually be available for public access from NASA web-services, similar to other NASA and FEWS NET supported land data assimilation (FLDAS) outputs.*

Some detailed comments:

**Comment #8**
Abstract: it should be made clear in there that the RZSM products are derived from the new NHyFAS. It reads like that they are not.

**Response:** Done. We have included the following sentence in the abstract:

*"For SA, this study documents the predictive capabilities of RZSM products from a recently developed NASA Hydrological Forecasting and Analysis System (NHyFAS)."*

**Comment #9**
lines 39-43: authors aren't obligated to show a graph (also that is helpful) but they should cite references.

**Response:** We have added the following figure, as Figure 2, showing that the percentage of income held by the bottom 10% and 20% of the population has not changed significantly in the region.

[Figure]

[Figure]

**Comment #10**

Line 137: the choice of RZSM as a hydrologic variable of interest makes sense but a rationale should be provided for it being the main (or indeed only) variable of interest in this study. What justifies not using other variables.

**Response:** Please see our response to your comment #5.

**Comment #11**

Figure 3, and commentary lines 230-250: this seems needlessly confusing. My understanding is that Fig 3 shows the correlation of crop yield with three monitoring-based products whereas the text touts the superiority of the forecasting-based product on all three as early as November. The latter, as well as one of the three monitoring products, includes RZSM, and the distinction is not always clear on first read.

Besides, back-and-forth with Figure 4 doesn't make the reading easy either. Could it be a good idea to 1) include the forecasting-based product on Figure 3 to provide a striking visual of why the proposed product is better, and 2) separate comment on Figure 3 from that of Figure 4.

**Response:** We apologize for the confusion. We have now revised the text discussing the results of Figure 3 and 4 (now Figure 5 and 6) to the following:

*First, we show in Figure 5 how detrended crop yield correlates (from early November to early March) with the monthly RZSM monitoring product relative to how it correlates with 3-monthly seasonal precipitation and air temperature. The results indicate that the monthly RZSM monitoring product generally correlates better with detrended crop yield than with the seasonal precipitation or air temperature, with the correlation reaching its peak by early March, when the Feb-RZSM monitoring product and December-February precipitation and temperature are available. Feb-RZSM still shows higher correlation than seasonal precipitation and temperature; however, the difference in correlation is not statistically significant.*

*Next, the correlation between detrended crop yield and February RZSM forecasts (based on ESP method and bias-corrected GEOS forecasts) initialized on November 1 (Fig. 6a) and January 1 (Fig. 6b) is analyzed. The correlation of the yield with GEOS-based February RZSM forecasts initialized on November 1 is 0.49, which is substantially higher than that of ESP-based RZSM forecasts (0.16), clearly demonstrating the added value of using GEOS-based climate forecasts. Similarly, the correlation of yield with the GEOS-based February RZSM forecasts initialized on January 1 is 0.45, higher than that of the ESP-based forecasts (0.30) at that time of the year. Moreover, the correlation of detrended crop yield with GEOS-based February RZSM forecasts initialized on November 1 (0.49) and January 1 (0.45) is higher than that with the RZSM monitoring product (Figure 5) at those times of the year (<0.1 in early November and*

*<0.4 in early January). Again, this highlights the value of using forecasts of Feb-RZSM through early January in supporting food insecurity early warning. Figure 6c shows that Feb-RZSM monitoring product, which is available in early March, has the highest correlation of 0.79 with the detrended crop yield.*

**Comment #12**
Line 266: why the lower tercile? Please justify

**Response:** We have now added the following text in the manuscript.

[revised manuscript text omitted]